# Genotype specific pathogenicity of hepatitis E virus at the human maternal-fetal interface

Jordi Gouilly [1], Qian Chen[1], Johan Siewiera[2], Géraldine Cartron[3], Claude Levy[4], Martine Dubois[5], Reem Al-Daccak[6], Jacques Izopet[1,5], Nabila Jabrane-Ferrat [1] & Hicham El Costa [1,5]

Hepatitis E virus (HEV) infection, particularly HEV genotype 1 (HEV-1), can result in fulminant hepatic failure and severe placental diseases, but mechanisms underlying genotype-specific pathogenicity are unclear and appropriate models are lacking. Here, we model HEV-1 infection ex vivo at the maternal-fetal interface using the *decidua basalis* and fetal placenta, and compare its effects to the less-pathogenic genotype 3 (HEV-3). We demonstrate that HEV-1 replicates more efficiently than HEV-3 both in tissue explants and stromal cells, produces more infectious progeny virions and causes severe tissue alterations. HEV-1 infection dysregulates the secretion of several soluble factors. These alterations to the cytokine microenvironment correlate with viral load and contribute to the tissue damage. Collectively, this study characterizes an ex vivo model for HEV infection and provides insights into HEV-1 pathogenesis during pregnancy that are linked to high viral replication, alteration of the local secretome and induction of tissue injuries.

[1] Centre of Pathophysiology Toulouse Purpan, INSERM U1043, CNRS UMR5282, Toulouse III University, 31024 Toulouse, France. [2] University of California San Francisco, School of Medicine, Laboratory of Medicine, San Francisco, CA, USA. [3] Service de Gynécologie-Obstétrique, Hôpital Paule de Viguier, Centre Hospitalier Universitaire, 31059 Toulouse, France. [4] Service de Gynécologie-Obstétrique, Clinique Sarrus-Teinturiers, 31300 Toulouse, France. [5] Laboratoire de Virologie, Institute of Federative Biology, Centre Hospitalier Universitaire, 31059 Toulouse, France. [6] INSERM UMRS976, Université Paris Diderot, Hôpital Saint-Louis, 75010 Paris, France. These authors contributed equally: Nabila Jabrane-Ferrat, Hicham El Costa. These authors share senior authorship: Nabila Jabrane-Ferrat, Hicham El Costa. Correspondence and requests for materials should be addressed to N.J.-F. (email: nabila.jabrane-ferrat@inserm.fr) or to H.E.C. (email: hicham.el-costa@inserm.fr)

epatitis E virus (HEV) is a single-stranded, positive polarity RNA virus that belongs to the *Hepeviridae* family. Human cases of hepatitis E are caused by *Orthohepevirus A* species, which comprises eight genotypes[1]. Genotypes 1 and 2 are obligate human pathogens transmitted by the fecal-oral route that has been associated with large outbreaks and epidemics in developing countries. Genotypes 3, 4, and 7 are responsible for sporadic cases of zoonotic hepatitis E, primarily in industrialized countries[2,3].

HEV infection is usually asymptomatic or causes acute self-limiting illness. In countries with poor sanitation, HEV-1 infection during pregnancy often results in fulminant hepatic failure (FHF, in 15–30% of cases) associated with severe placental diseases, including eclampsia, hemorrhage, membrane rupture, spontaneous abortion, and stillbirths[2,4]. These fatal outcomes resemble those reported for other harmful pathogens during pregnancy and can be attributed to dysfunctions at the maternal-fetal interface composed of the pregnant endometrium lining *or decidua basalis* (decidua), and fetal placenta[5–7]. In western countries where genotype 3 (HEV-3) prevails, HEV infection is rather harmless during pregnancy[8–10].

Significant effort has been made toward understanding HEV-1-induced FHF during pregnancy. However, the causal relationship between infection and placental dysfunction remains elusive. The lack of appropriate in vivo and in vitro experimental models and the difficulty to propagate the virus in vitro hampered the understanding of this genotype-specific pathogenesis in pregnant women. Although case reports and longitudinal studies using peripheral blood samples have incriminated viral load, immune response, hormones, and few signaling pathways in the observed adverse pregnancy outcomes[4,11], peripheral responses do not accurately reflect the events occurring at the maternal-fetal interface[12]. To our knowledge, only a single study identified viral components in placental tissue from HEV-1-infected women, suggesting the placenta as an extra-hepatic site for viral replication[13].

To provide insights into the genotype-specific pathogenicity of HEV during pregnancy, we ex vivo modeled the pathological HEV-1 and less-pathological HEV-3 infection at the maternal-fetal interface using organ culture from the maternal decidua and fetal placenta. Whereas both HEV genotypes are able to infect the maternal-fetal interface, HEV-1 replicates more efficiently in decidua and placenta tissue explants as well as in primary fibroblast-like stromal cells isolated from both tissues, produces higher amounts of infectious progeny virions and causes severe morphological alterations. Furthermore, viral replication correlates with pronounced alterations in the cytokine, chemokine, and growth factor networks at the maternal-fetal interface resulting in an exacerbated tissue injury. Collectively, the genotype-specific fatal outcomes during pregnancy are likely linked to efficient viral replication and a dysregulated local secretome.

## Results

**Maternal-fetal interface supports high HEV-1 replication**. We ex vivo modeled HEV infections at the maternal-fetal interface using organ cultures of the maternal decidua and fetal placenta from elective pregnancy terminations and examined their susceptibility to clinical strains of HEV-1 and HEV-3 isolated at the acute phase of infection from the feces of a traveler returning from India and an autochthone infected patient, respectively. Quantitative RT-PCR analysis revealed a substantial replication of HEV-1 that reached a plateau within 2 days in the placenta and 5 days in the decidua (Fig. 1a, b). Although, following similar trend, HEV-3 replication was significantly lower in both tissues.

HEV replication is highly sensitive to anti-viral drugs such as ribavirin (RBV) both in vitro and in clinical settings[2,14]. To further corroborate our observation that HEV replicates at the maternal-fetal interface, we treated infected tissues with an optimal dose of 50 μM of RBV and monitored viral replication over time. Treatment with RBV resulted in strong inhibition of HEV-1 and HEV-3 replication both in the decidual and placental tissue explants (Supplementary Fig. 1). The lack of viral replication in tissue explants exposed to UV-irradiated virions further strengthens the active replication of HEV at the maternal-fetal interface. The fact that RBV treatment did not completely abrogate the viral replication in the placenta organ cultures is probably due to the complexity of the tissue architecture. Nonetheless, our findings demonstrate that the detection of viral genome in the culture supernatants is reminiscent of active viral replication.

To ascertain the tropism of HEV-1 during pregnancy, we then performed in situ hybridization (ISH) on the infected tissue explants using a set of probes covering the whole-HEV genome (Fig. 1c, d). Hybridization with HEV probes resulted in a higher amount of dot-like signals in HEV-1 infected samples compared to HEV-3. In addition, HEV-1 positive cells showed a spatial clustered signal distribution indicative of replication foci (Fig. 1c). Compared to HEV-3, HEV-1-related signal tends to be more intense (Fig. 1c). Nonetheless, quantitative analysis using large microscopy fields revealed a 2 to 3-fold higher number of cells infected with HEV-1 than HEV-3 in both tissues (Fig. 1d).

Taken together, these experiments demonstrate that the maternal-fetal interface supports HEV replication with a better tropism for HEV-1.

**HEV-1 infection is associated with tissue injury**. Lessons from the TORCH (Toxoplasma, Others, Rubella virus, Cytomegalovirus, and Herpes simplex virus) infections have revealed that viral replication at the maternal-fetal interface is associated with tissue damage and adverse pregnancy outcome[15,16]. Furthermore, apoptotic and necrotic features were previously reported in liver biopsies from HEV infected patients[17,18]. Accordingly, we assessed whether HEV infection impairs the morphology of the maternal-fetal interface. We first evaluated the programmed cell death by Terminal deoxynucleotidyl transferase dUTP nick end labelling (TUNEL) staining of tissue sections 5 days post infection (Fig. 2a, b). Mock-infected tissues showed a sparse to no TUNEL staining; whereas a significant increase in the proportion of TUNEL positive cells in HEV-1 infected decidua and placenta tissue explants was observed (Fig. 2a, b). Apoptotic cells were depicted both in the outer syncytiotrophoblast and the inner cytotrophoblast layers as well as in the villus core of the placenta. In contrast to HEV-1, HEV-3 infection led to scattered apoptotic cells in both the decidua and placenta (Fig. 2a, b).

We next analyzed whether HEV infection impairs the architecture of the maternal-fetal interface. Hematoxylin-Eosin (HE) staining demonstrates that HEV-1 infection induces significant injury to the decidua and placenta as evidenced by the necrotic features, such as pyknosis, karyorrhexis, and karyolysis (Fig. 2c, d). Furthermore, HEV-1-infected placenta displayed a drastic disruption of the trophoblastic layer (Fig. 2d). The effects of HEV-3 infection were milder in both tissues (Fig. 2c, d).

Taken together, our findings demonstrate that HEV-1 infection induces significant increase in tissue injury at the maternal-fetal interface that might account for the morbid pregnancy complications reported during pregnancy.

**HEV-1 skews the tissue secretome and promotes tissue damage**. The soluble microenvironment of the uterine mucosa,

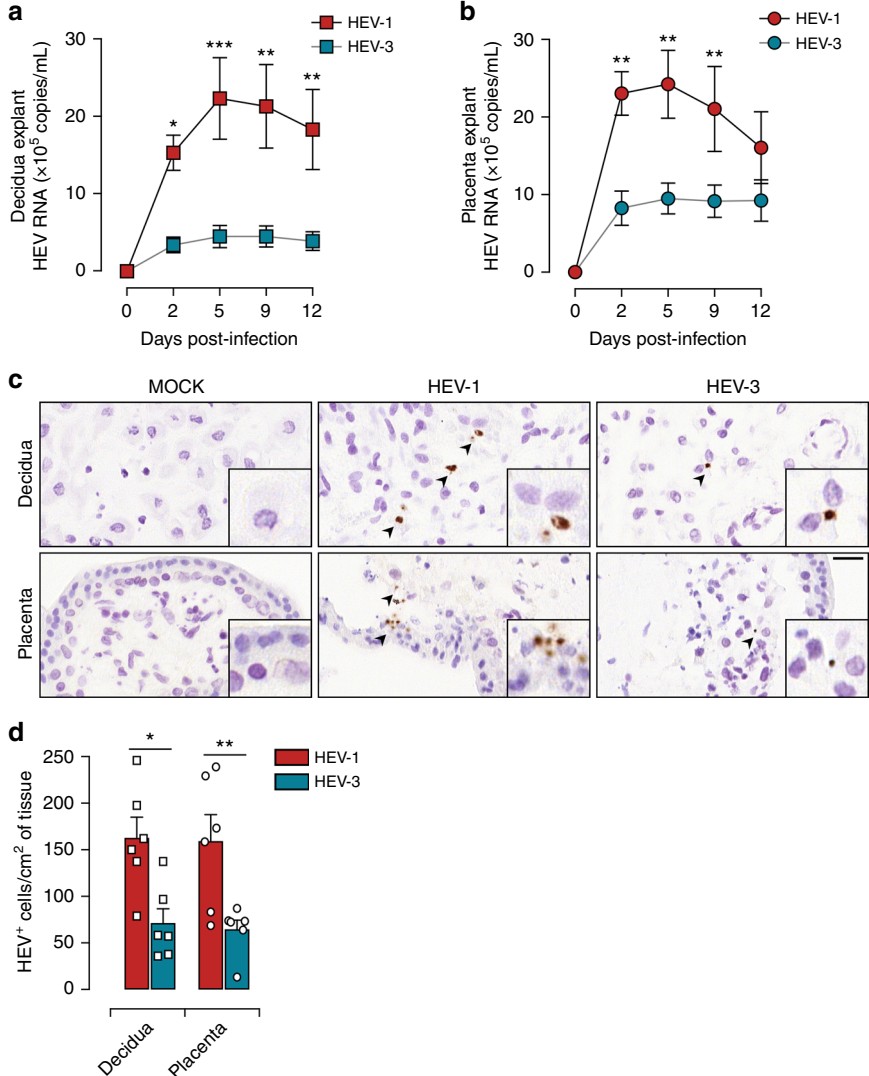

**Fig. 1** HEV-1 replicates efficiently in decidual and placental tissues. **a**, **b** Kinetics of HEV virus production from explants established from the decidua **a** and placenta **b**, infected with HEV-1 (red), or HEV-3 (cyan). RNA levels were measured in tissue culture supernatants by RT-qPCR. **c**, **d** Histological analyses of tissue sections stained by HEV in situ hybridization (ISH) and prepared from mock, HEV-1, or HEV-3 infected explants 5 days post infection. **c** Representative field of view of explant material derived from the decidua (upper panel) and placenta (lower panel). Arrowheads point to HEV positive cells (in brown) and boxes represent enlarged areas with characteristic staining patterns. Scale bar, 20 μm. **d** Bar graph illustrating the number of HEV positive cells per cm$^2$ of tissue determined by ISH staining in HEV-1 (red) or HEV-3 (cyan) infected tissue explants. Data represent mean values ± S.E.M. of six independent donors. * denotes a statistical comparison between HEV-1 and HEV-3 infected tissues. *$P < 0.05$; **$P < 0.01$; ***$P < 0.001$ by two-way ANOVA with Bonferroni post hoc test **a**, **b** and paired $t$-test **d**

within first semester of pregnancy, is mandatory not only for embryo implantation but also for the maternal-fetal tolerance. Any alteration to the local secretome can provoke pregnancy-related disorders[19,20]. Therefore, we assessed the impact of HEV-1 and HEV-3 infection on the maternal-fetal interface secretory profile. Based on their respective roles during pregnancy[19–22], the levels of selected cytokines (IL-6, IL-15, IL-18, TNF-α, and sICAM-1), chemokines (CCL-3, CCL-4, CCL-5, and CXCL-10), growth factors (G-CSF, GM-CSF, and VEGF-A), and metalloproteinases (MMP-2 and MMP-9) were quantified in tissue culture supernatants 2 days post infection (Fig. 3).

We observed three secretion profiles upon infection of decidual and placental tissues with HEV. The first profile includes factors that are not affected by infection, such as G-CSF, VEGF-A, MMP-2, and MMP-9 in the decidua (Fig. 3a) and VEGF-A, MMP-2 in the placenta (Fig. 3b). The second profile comprises factors that are increased by infection independently of the viral

genotype including sICAM-1 and GM-CSF in both tissues (Fig. 3a, b), or G-CSF and MMP-9 only in the placenta (Fig. 3b). The last profile consists of pro-inflammatory factors (IL-6, CCL-3, CCL-4, and CXCL-10) that are differentially regulated by HEV-1 and HEV-3 infection. Compared to HEV-3, IL-6, CCL-3, and CCL-4 secretion is markedly enhanced by HEV-1 infection, while CXCL-10 is significantly depressed in both tissues. Regardless of the infection, TNF-α, IL-15, IL-18, and CCL-5 were absent or barely detected.

We next investigated the relationship between the level of the differentially altered factors and HEV-1 or HEV-3 replication in decidual and placental tissue. The secretion levels of IL-6, CCL-3, CCL-4, CXCL-10, and HEV-1 load in decidual explants were highly correlated with a $0.0001 \le P$-value $\le 0.003$ and a $R$-value $\ge 0.80$ using Spearman's ranked correlation test (Fig. 3c). HEV-1 load was also highly correlated with CCL-3, CCL-4, CXCL-10 levels in the placenta with a $0.047 \le P$-value $\le 0.0001$ and a $R$-value $\ge 0.82$ using

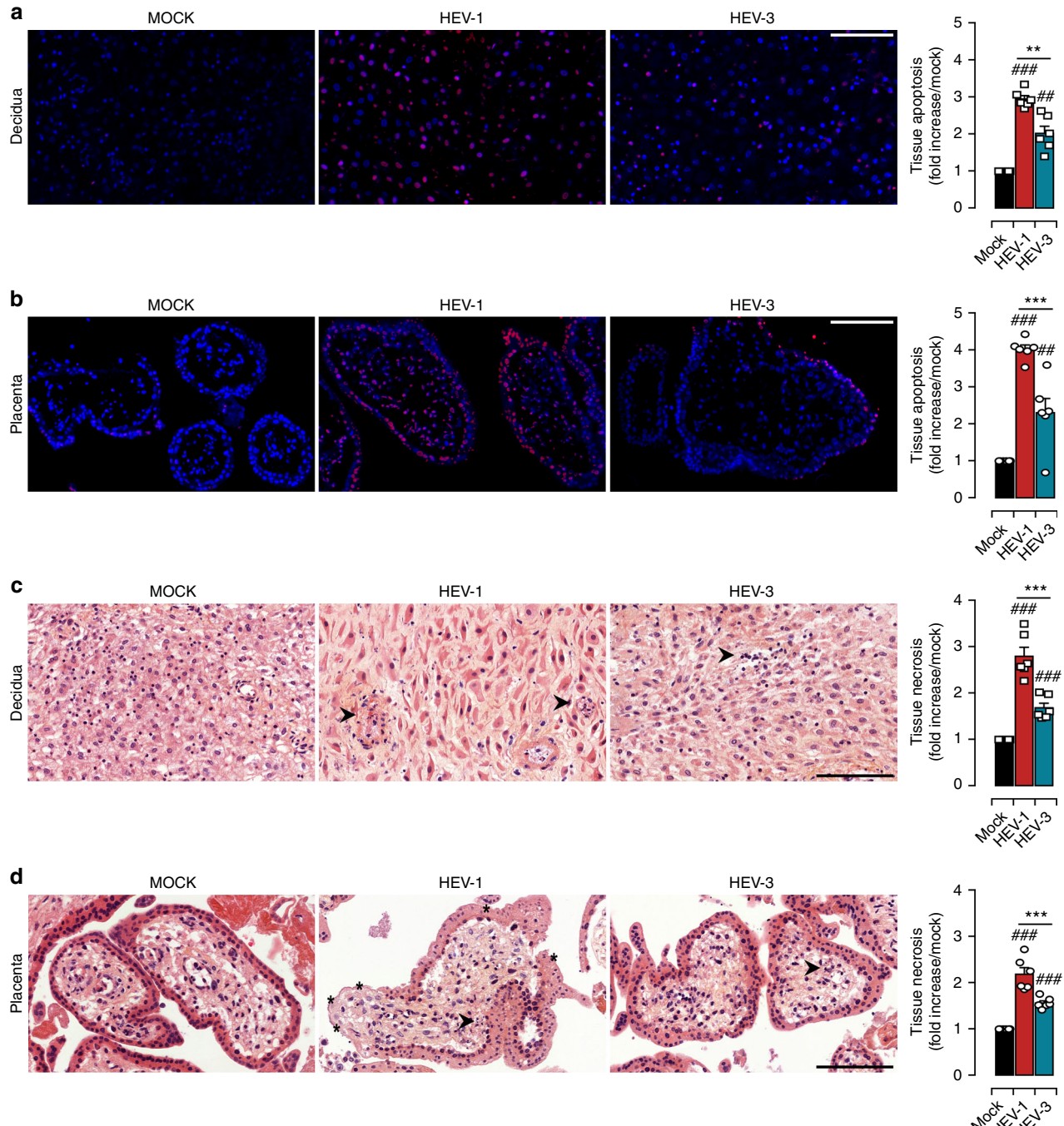

**Fig. 2** HEV-1 causes severe injury in decidual and placental tissues. Histological analyses of apoptosis (TUNEL staining, **a**, **b**) and necrosis (H&E staining, **c**, **d**) in explants sections prepared from mock, HEV-1, or HEV-3 infected tissues, 5 days post infection. **a**, **b** Representative large field of view of TUNEL stained sections prepared from the decidua **a** and placenta **b**. Staining indicates the apoptotic cells (red) and nuclei (blue). Scale bar, 100 μm. Bar graph illustrates the increase of tissue apoptosis in HEV-1 (red) or HEV-3 (cyan) infected tissue explants, compared to levels detected in mock-infected tissue explants (black) and represented as fold increase. **c**, **d** Representative large field of view of H&E stained sections prepared from the decidua **c** and placenta **d**. Arrowheads point to necrotic zones with nuclear changes illustrated by pyknosis, karyorrhexis, and karyolysis. Stars indicate an injured syncytiotrophoblast layer. Scale bar, 100 μm. Bar graph illustrates the increase of tissue necrosis in HEV-1 (red) or HEV-3 (cyan) infected tissue explants, compared to levels detected in mock-infected tissue explants (black) and represented as fold increase. Data represent mean values ± S.E.M. of six independent donors. * denotes a statistical comparison between HEV-1 and HEV-3 infected tissues and # represents a statistical comparison between mock and HEV-1 or HEV-3 infected tissues. **/##$P < 0.01$; ***/###$P < 0.001$ by repeated measures ANOVA with Tukey post hoc test

Spearman's ranked correlation test (Fig. 3d). Although, it reached significance, the effect of HEV-1 replication on the IL-6 placental production was very low. Unlike HEV-1, only moderate correlations were observed between HEV-3 and cytokine production both in the decidua and placenta samples (Supplementary Fig. 2).

To further visualize the global pattern of soluble factor secretion during HEV-1 and HEV-3 infection, we included IL-6, CCL-3, CCL-4, and CXCL-10 decidual and placental levels in a principal component analysis (Fig. 3e). HEV-1 and HEV-3 infected samples naturally clustered into two distinct groups with

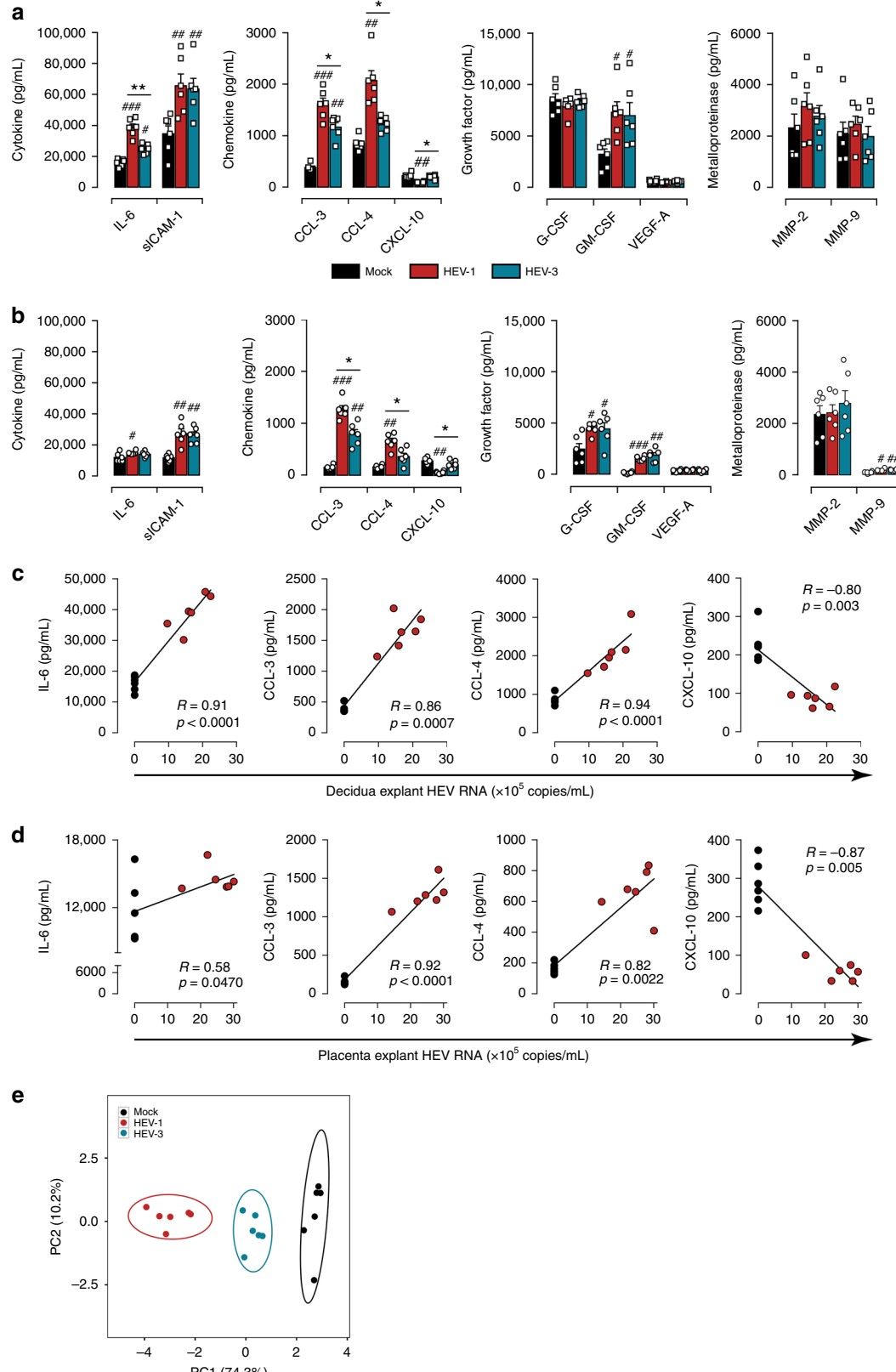

HEV-1 at the far end of the mock, while HEV-3 clustered at an intermediate position. Thus, PCA analysis reveals a genotype-specific cytokine signature that discriminates HEV-1 from HEV-3 and may contribute to the differential viral pathogenicity at the maternal-fetal interface.

To determine the impact of the changes in the cytokine environment, we challenged fresh decidua and placenta organ cultures with UV-irradiated conditioned medium (UV-CM) harvested from their respective HEV-1- or HEV-3-infected explant cultures. The TUNEL assay (Fig. 4a, b) and HE staining

**Fig. 3** HEV-1 infection of decidual and placental tissues skews their secretory function. **a**, **b** Cytokine, chemokine, growth factor, and metalloproteinase secretion from explants established from either decidua **a** or placenta **b**, measured by multiplex assay in culture supernatants 2 days after infection with HEV-1 (red) or HEV-3 (cyan). Black bars represent mock-infected tissues. **c**, **d** Correlation between IL-6, CCL-3, CCL-4, or CXCL-10 secretion and viral production in the decidua **c** and placenta **d** 2 days after HEV-1 infection. Black and red points represent mock-, and HEV-1-infected tissues, respectively. The Spearman's rank correlation test *P*-value and R coefficient are indicated in each graph. **e** Principal component analysis (PCA) of IL-6, CCL-3, CCL-4, and CXCL-10 secretion in the decidua and placenta, 2 days after mock (black), HEV-1 (red), or HEV-3 (cyan) infection. Values were centered, unit variance scaling was applied to rows and single value decomposition with imputation was used to calculate principal components. *X* and *Y* axis show PC1 and PC2 that explain 74.3% and 10.2% of the total variance, respectively. Prediction ellipses are such that, with probability of 0.95, a new observation from the same group will fall inside the ellipse. Data represent mean values ± S.E.M. of six independent donors. * denotes a statistical comparison between HEV-1 and HEV-3 infected tissue. # represents a statistical comparison between mock and HEV-1 or HEV-3 infected tissues. */#*P* < 0.05; **/##*P* < 0.01; ###*P* < 0.001 by repeated measures ANOVA with Tukey post hoc test

of histological sections (Fig. 4c, d) revealed that the tissue injury is significantly more prominent with UV-CM obtained from HEV-1-infected samples for both the decidua and placenta. Nonetheless, the tissue disruption in the presence of UV-CM is less prominent than with the original inoculum (Fig. 2) suggesting that HEV-induced damage was at least partially due to alterations in the cytokine/chemokine networks.

Taken together, our data demonstrate that HEV-1 infection has higher capacity than HEV-3 to alter the secretome at the maternal-fetal interface further contributing to HEV-1 associated tissue injury.

**HEV-1 replication impairs the type III interferon secretion.** Innate sensing of invading pathogens has been clearly associated with the production of interferons (IFNs) that can impede the replication of a broad spectrum of viruses[23–27]. Nonetheless, at mid-gestation, the human placenta produces large amounts of type III IFNs even in the absence of threats. This constitutive release of IFNs correlated with expression of interferon-stimulated genes (ISGs) have been suggested as a persistent defense mechanism during the second trimester of pregnancy[28]. In order to shed light on the mechanisms that underlie the discrepancies between HEV-1 and HEV-3 replication at the maternal-fetal interface, we assessed the secretion profiles of type I (IFN-α2 and -β), type II (IFN-γ) and type III (IFN-λ1, -λ2/3) IFNs in tissue culture supernatants 2 days post infection (Fig. 5). Regardless of the infection, both the decidua and the placenta produced low levels of IFN-α2, IFN-β and IFN-γ (Fig. 5a, b). However, HEV-1 infection significantly impaired the production of IFN-λ1 and IFN-λ2/3 in the decidua, and IFN-λ2/3 in the placental explants, while HEV-3 infection has no impact on type III IFN secretion (Fig. 5a, b). Correlation analyses demonstrated that the production of IFN-λ1 and IFN-λ2/3 is negatively correlated with HEV-1 load in the decidua (*p* ≤ 0.0087 by Spearman's ranked correlation test, Fig. 5c), while correlation is observed only for IFN-λ2/3 in the placental tissues (*p* = 0.0096 by Spearman's ranked correlation test, Fig. 5d).

To ascertain the role of type III IFNs in HEV infection, we treated infected tissues with either IFN-λ1 or IFN-λ2 and monitored viral replication over time. Quantitative RT-PCR analysis revealed that both cytokines significantly inhibit HEV-1 replication in the decidual tissue explants (Fig. 5e). However, the effect of these cytokines was less prominent in the placental settings and inhibition was observed only with IFN-λ2 treatment (Fig. 5f).

Taken together, these findings suggest that HEV-1 antagonizes type III IFN production to sustain its efficient replication at the maternal-fetal interface.

**HEV-1 targets stromal cells at the maternal-fetal interface.** Stromal cells contribute actively to the maintenance of tissue homeostasis by supporting the tissue vascular remodeling and the

development of the fetal placenta, which are mandatory for a successful pregnancy[29]. Several pathogens that threaten the pregnancy replicate within these cells impacting their proper function[22,29–31]. Therefore, we next assessed whether fibroblast-like stromal cells (stroma) derived from decidual and placental tissues could support HEV replication (Fig. 6). We infected freshly isolated decidual and placental stromal cells with either HEV-1 or HEV-3 and monitored viral replication over time. In parallel, we used the prototypic human hepatocellular carcinoma HepG2/C3A cell line as a control (Supplementary Fig. 3). Quantitative RT-PCR analysis demonstrated that the replication of HEV-1 was significantly higher than HEV-3 after 7 days of infection, reaching 4.6-fold and 2.4-fold increase in decidual and placental stromal cells respectively (Fig. 6a, b). The presence of the ORF2 viral capsid protein in both stromal cell types further confirmed the higher infection of HEV-1 (Fig. 6c, d). In contrast to stromal cells, the HepG2/C3A cell line showed efficient replication only for HEV-3 as demonstrated by quantitative RT-PCR analysis and ORF2 immunostaining (Supplementary Fig. 3). In agreement with previous data[32], microscopic analysis did not show any cytopathic effects of HEV-1 or HEV-3 in these primary stroma cells.

Collectively, our data identify the maternal-fetal stromal cells as targets for HEV infection that are more likely to support HEV-1 replication rather than HEV-3.

**HEV-1 alters the secretome of primary stromal cells.** We next investigated whether HEV infection impairs the dynamic secretory function of stroma cells[19,20,22,29]. Two days post infection, supernatants were collected from HEV-1, HEV-3, or mock-infected decidual and placental stromal cell cultures, and soluble factors were subsequently quantified (Fig. 7). Compared to HEV-3, HEV-1 infection resulted in significant increase of several pro-inflammatory mediators, including IL-6, sICAM-1, CCL-3, CCL-4, G-CSF, and GM-CSF in decidual cells (Fig. 7a). Infection of placental cells induced lesser amounts of these mediators with significantly higher secretion of IL-6 and sICAM-1 in HEV-1 infected cultures (Fig. 7b). Irrespective of HEV genotype, VEGF-A was increased only in placental stroma and the production of metalloproteinases was not affected in both cell types (Fig. 7a, b).

Taken together, these results demonstrate that HEV-1, and to a lesser extent HEV-3, skews the cytokine, chemokine, and growth factor secretory profile in both the decidual and placental stroma, corroborating the observations made with tissue explants. Thus, the HEV-1 infection of decidual and placental stroma cells may not only favor viral dissemination but also impair their tissue-support functions and underlie some of the adverse pregnancy outcomes associated with this HEV genotype.

**HEV-1 and HEV-3 progeny display differential infectivity.** To provide further insights into the differential pathogenicity of

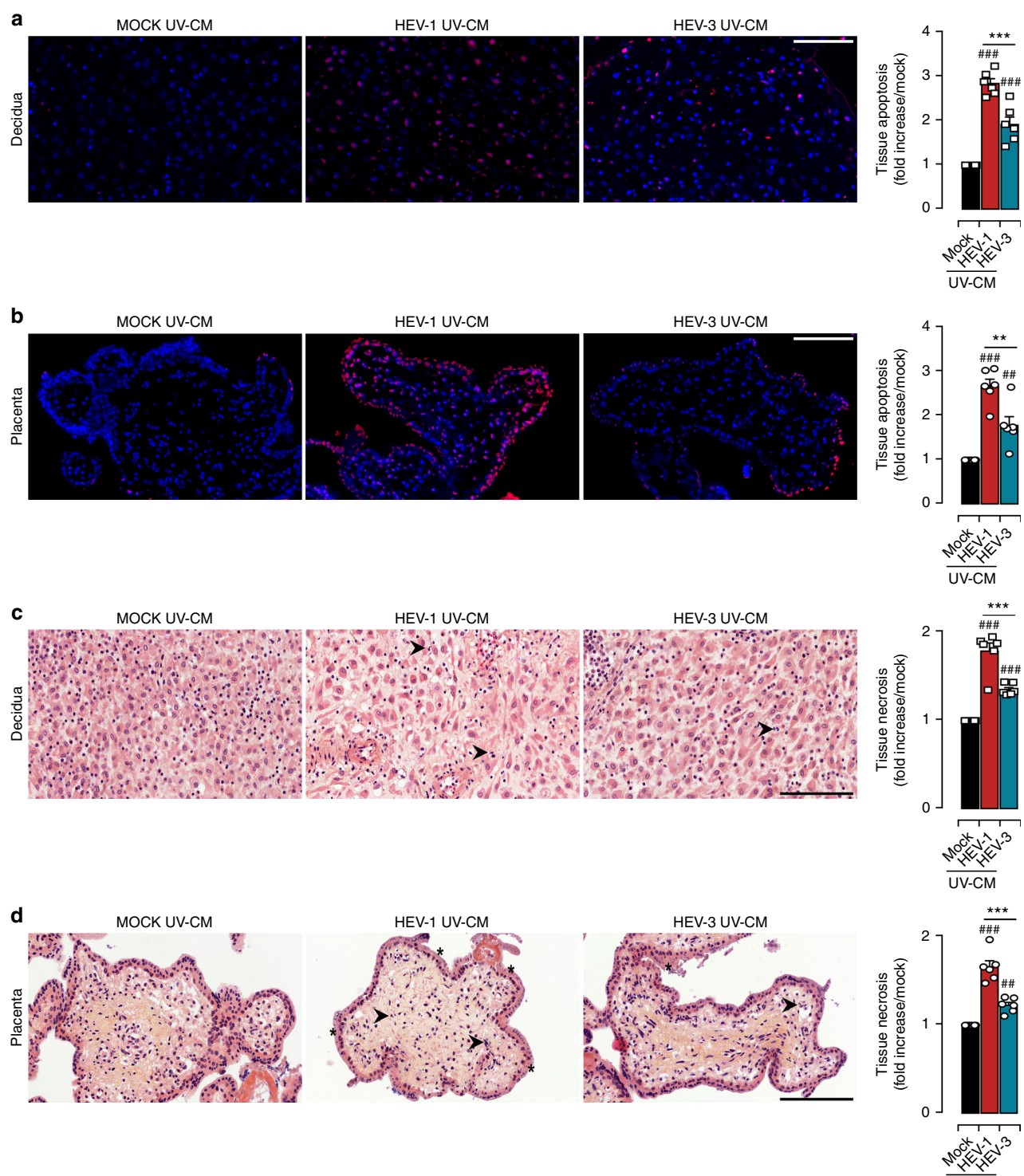

**Fig. 4** HEV-1 altered secretome contributes to tissue damage. Histological analyses of apoptosis (TUNEL staining, **a**, **b**) and necrosis (H&E staining, **c**, **d**) and in explants sections prepared from tissues challenged for 5 days with either Mock, HEV-1, or HEV-3 UV-irradiated conditioned media (CM). **a**, **b** Representative large field of view of TUNEL stained sections prepared from the decidua **a** and placenta **b**. Staining indicates the apoptotic cells (red) and nuclei (blue). Scale bar, 100 μm. Bar graph illustrates the increase of tissue apoptosis in explants challenged for 5 days with either HEV-1 (red) or HEV-3 (cyan) UV-CM. Results are normalized to data obtained using UV-CM harvested from mock-infected explants (black) and represented as fold increase. **c**, **d** Representative large field of view of H&E stained sections prepared from the decidua **c** and placenta **d**. Arrowheads point to necrotic zones with nuclear changes illustrated by pyknosis, karyorrhexis, and karyolysis. Stars indicate an injured syncytiotrophoblast layer. Scale bar, 100 μm. Bar graph illustrates the increase of tissue necrosis in explants challenged for 5 days with either HEV-1 (red) or HEV-3 (cyan) UV-CM. Results are normalized to data obtained using UV-CM harvested from mock-infected explants (black) and represented as fold increase. Data represent mean values ± S.E.M. of six independent donors. * denotes a statistical comparison between HEV-1 and HEV-3 infected tissues and # represents a statistical comparison between mock and HEV-1 or HEV-3 infected tissues. **/##$P < 0.01$; ***/###$P < 0.001$ by repeated measures ANOVA with Tukey post hoc test

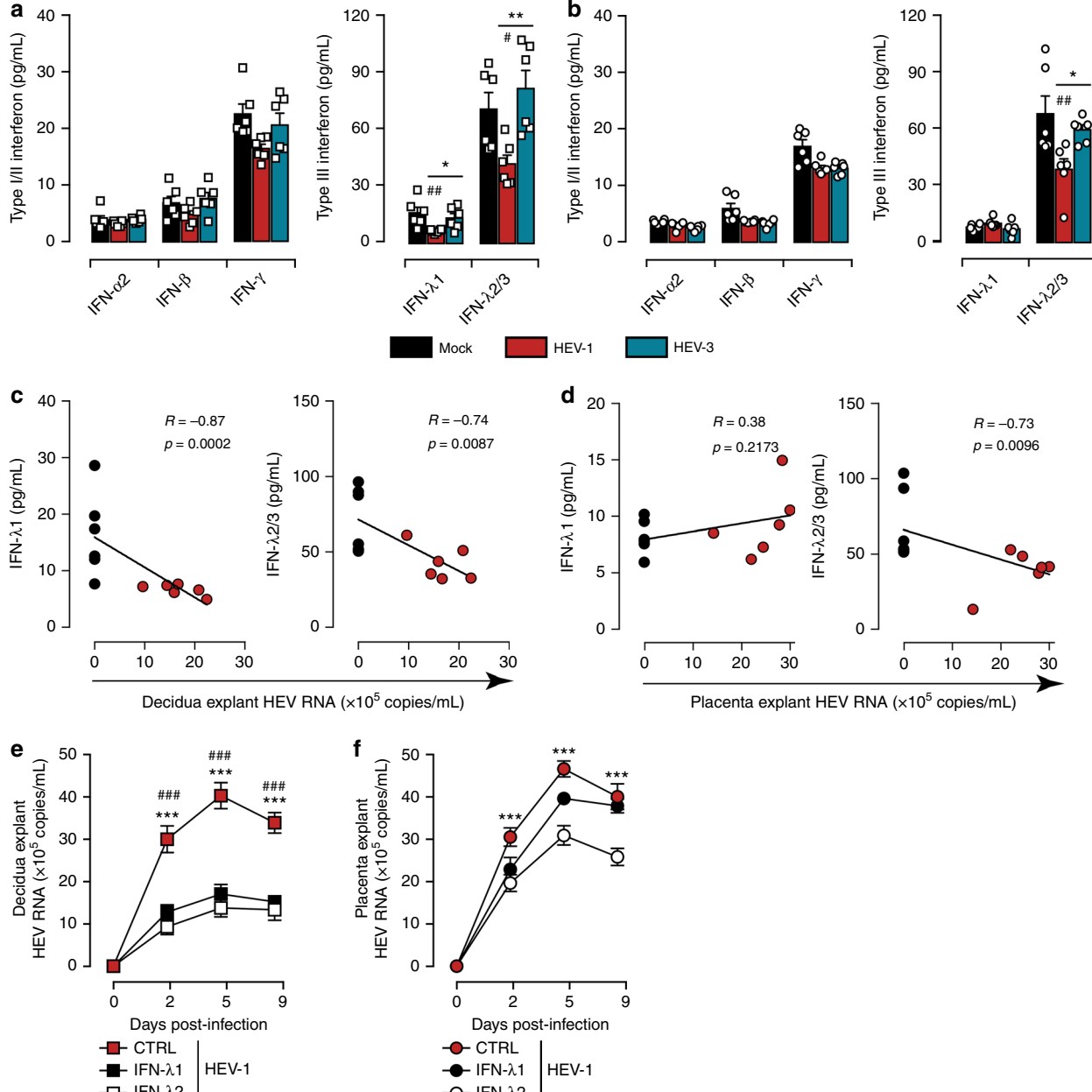

**Fig. 5** HEV-1 replication is associated with a decrease in type III interferon levels. **a**, **b** Type I/II and type III interferon secretion from explants established from either decidua **a** or placenta **b**, measured by cytometric bead array in culture supernatants 2 days after infection with HEV-1 (red) or HEV-3 (cyan). Black bars represent mock-infected tissues. Data represent mean values ± S.E.M. of six independent donors. * denotes a statistical comparisons made between HEV-1 and HEV-3 infected tissue, and # represents a statistical comparison between mock-infected and HEV-1 or HEV-3 infected tissue. */#$P < 0.05$; **/##$P < 0.01$. **c**, **d** Correlation between IFN-λ1, or IFN-λ2/3 secretion and viral production in the decidua **c** and placenta **d** 2 days after HEV-1 infection. Black and red points represent mock-, and HEV-1-infected tissues, respectively. The Spearman's rank correlation test $P$-value and $R$-coefficient are indicated in each graph. **e**, **f** Kinetics of HEV virus production from explants established from the decidua **e** and placenta **f**, infected with HEV-1. After infection, tissues were left untreated (CTRL, in red), treated with IFN-λ1 (100 ng/mL, in black) or treated with IFN-λ2 (100 ng/mL, in white). RNA levels were then measured in tissue culture supernatants by RT-qPCR. Data represent means values ± S.E.M. of three independent donors. * denotes a statistical comparison between CTRL- and IFN-λ2-treated infected tissues. # denotes a statistical comparison between CTRL- and IFN-λ1-treated infected tissues. ***/###$P < 0.001$ by repeated measures ANOVA with Tukey post hoc test **a**, **b** and two-way ANOVA with Bonferroni post hoc test **e**, **f**

HEV-1 and HEV-3 at the maternal-fetal interface, we assessed the infectivity of HEV-1 and HEV-3 progeny virions. Equivalent copy number of viral RNA recovered from HEV-1 and HEV-3 infected decidual and placental explant supernatants were used to infect primary stromal cells or HepG2/C3A cell line. Data depicted in Fig. 8 demonstrate that supernatants recovered from infected tissues contain infectious progeny virions that can replicate in

both type of cells. Compared to HEV-3, HEV-1 virions replicated at a significantly higher rate in decidual and placental stroma cells (Fig. 8a, b). In contrast, HEV-3 virions replicated more efficiently in HepG2/C3A cells (Fig. 8c, d). Thus, HEV-1 infection of the maternal-fetal tissues generates infectious progeny virions that preserve the tropism of the original inoculum and replicate more efficiently than HEV-3 in both decidual and placental stroma

cells. The HEV-1 progeny virions released at the maternal-fetal interface could then easily spread to neighboring cells, thereby promoting viral dissemination in utero and might explain the underlying pathogenicity of HEV-1 rather than HEV-3 during pregnancy.

## Discussion

The mechanisms underlying HEV genotype-specific severity during pregnancy and viral transmission to the fetus remain poorly understood. Using an ex vivo model of first trimester maternal-fetal interface, we provide here the first evidence that the decidua and the placenta are prone to HEV-1 infection rather than HEV-3, resulting in the generation of infectious progeny virions.

Being highly irrigated, the maternal-fetal interface is exposed to viral particles present in circulating blood after ingestion of infected components or release of progeny virions from infected hepatocytes[33,34]. The presence of high viral loads in sera from pregnant women with FHF and experiencing placental disease is in favor of this hematogenous spread[35]. Our findings advance the state-of-the-art by highlighting the interplay between the decidua and the placenta at the maternal-fetal interface in association with the preponderance of stroma cells as a vertical transmission mechanism of HEV similar to the TORCH pathogens[5,6,19,30,36].

Genotype-dependent factors may favor HEV-1 replication in decidual and placental tissues. The newly discovered ORF4 protein encoded by the HEV-1 genome could therefore recruit other viral proteins and/or host-related factors to promote HEV-1 replication[37]. The expression of the ORF1 and ORF3 viral proteins can also impair the host innate immune response through blockade of the IFN or retinoic-acid-inducible gene-1 (RIG-1)-like signaling pathways resulting in viral escape[38,39]. Several reports have highlighted differences in the production of IFNs as well as the subsequent induction of ISGs[23–27,40]. Both the HEV strain and experimental settings could probably contribute to these discrepancies. Herein, neither HEV-1 nor HEV-3-induced type I and type II IFN secretion at the maternal-fetal interface. However, HEV-1 viral load was negatively correlated to the expression of type III IFNs and CXCL-10. This significant decrease of IFN-λ, associated with the inhibition of HEV-1 replication by type III IFNs, suggest that HEV-1 subverts the IFN signaling pathway to replicate efficiently. The moderate inhibition of HEV-1 by IFN-λ2 in placental tissue might be inherent to the tissue architecture and factors, the induction of IFN resistant ISGs or persistent activation of the JAK/STAT signaling[25].

Our findings highlight also a distinct pathogenicity between HEV genotypes at the maternal-fetal interface and provides a plausible explanation for the discrepant outcomes between HEV-1 and HEV-3 infection during pregnancy. We clearly demonstrate that HEV-1 is associated with increased apoptosis and necrosis at the maternal-fetal interface with alterations of the placental barrier architecture. HEV-associated tissue apoptosis and necrosis have been previously reported in liver biopsies of infected patients and animal models[17,18,41]. While, the origin of this tissue injury remains elusive a recent study suggested that it might involve mitochondrial damage and/or activation of the caspase family members[41]. Our results revealed also that HEV-1 infection alters the secretion profile of decidual/placental tissues and cells. These findings are consistent with data correlating the increase of several pro-inflammatory factors in the peripheral blood from HEV-1 infected women with adverse pregnancy outcomes[42]. Under steady-state conditions, the secretome within the female reproductive tract and gestational tissue is finely tuned to support embryo implantation, placental development, and fetal immune tolerance. Conversely, an uncontrolled, prolonged, or

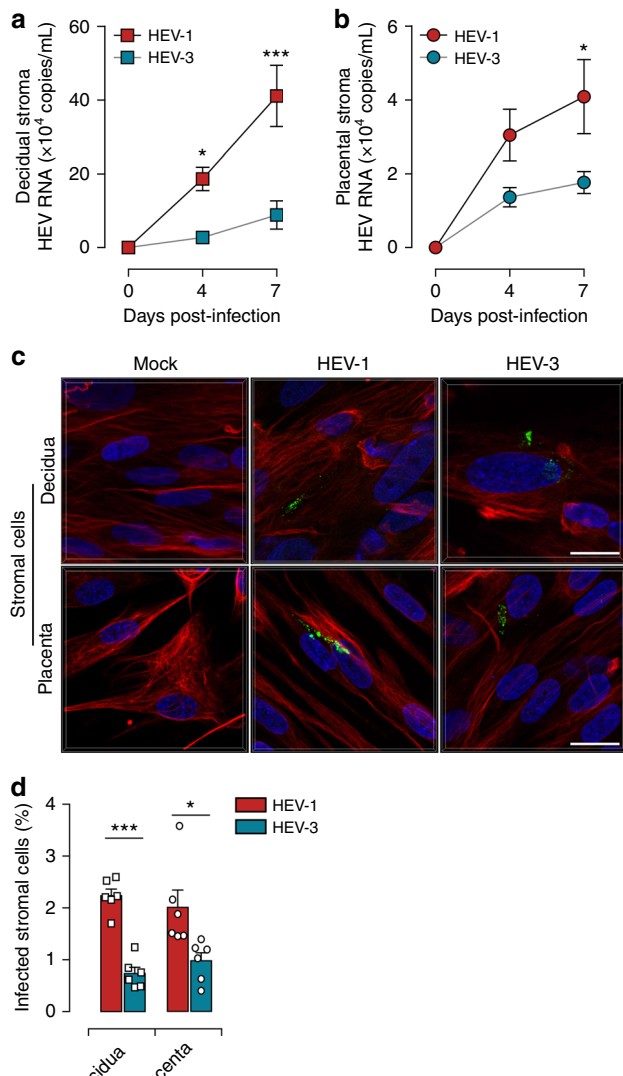

**Fig. 6** Primary decidual and placental stromal cells are targets of HEV-1 infection at the maternal-fetal interface. **a**, **b** Kinetics of viral RNA production in stromal cells derived from the **a** decidua or **b** placenta, infected with either HEV-1 (red) or HEV-3 (cyan). Virus production was determined by RT-qPCR in culture supernatants. **c** Representative images of stromal cells derived from the decidua (upper panel) or placenta (lower panel), 7 days after mock, HEV-1, or HEV-3 infection. 3D-reconstituted maximum intensity projections are shown, generated using the Imaris software. Staining indicates the ORF2 viral capsid protein (green), vimentin (red), and nuclei (blue). Scale bar, 20 μm. **d** Bar graph illustrating the percentage of infected stromal cells derived from the decidua or placenta 7 days after HEV-1 (red) or HEV-3 (cyan) infection and determined by ORF2 staining. Data represent mean values ± S.E.M. of six independent donors. * denotes a statistical comparison made between HEV-1 and HEV-3 infected cells. *$P < 0.05$; ***$P < 0.001$ by two-way ANOVA with Bonferroni post hoc test **a**, **b** and paired $t$-test **d**

excessive mediator secretion in response to viral infection has previously been associated with tissue injury and adverse pregnancy outcomes[19–22,43]. Indeed, a pro-inflammatory microenvironment may lead to the recruitment and activation/differentiation of pro-inflammatory cells and/or additional viral targets[44,45]. Consequently, a positive feedback loop can be established and exacerbates the local response, further amplifying tissue damage and viral spread. By modulating cytokine and

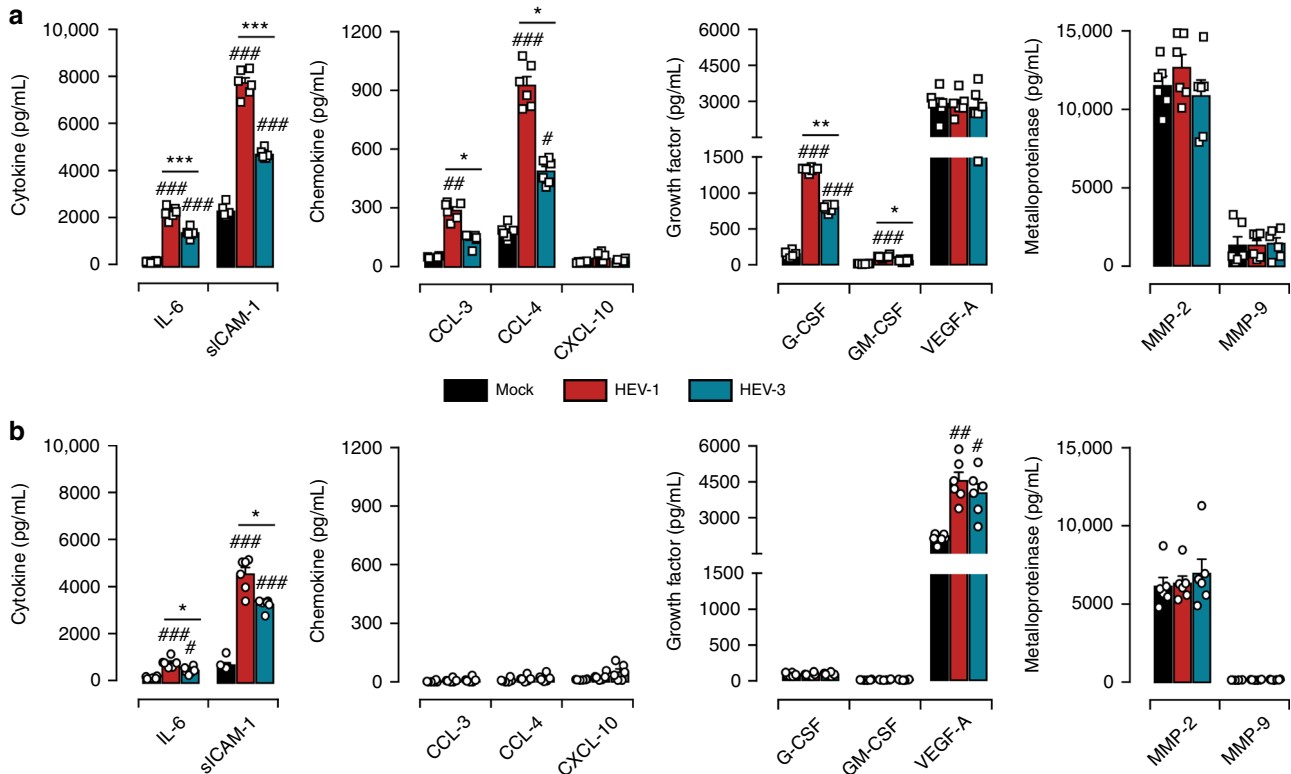

**Fig. 7** HEV-1 alters the secretory functions of both decidual and placental stromal cells. **a, b** Cytokine, chemokine, growth factor, and metalloproteinase secretion in decidual **a** and placental **b** stromal cell supernatants, measured by multiplex assay two days after mock (black), HEV-1 (red), or HEV-3 (cyan) infection. Data represent mean values ± S.E.M. of six independent donors. * denotes a statistical comparisons made between HEV-1 and HEV-3 infected tissue, and # represents a statistical comparison between mock-infected and HEV-1 or HEV-3 infected tissue. */#$P < 0.05$; **/##$P < 0.01$; ***/###$P < 0.001$ by repeated measures ANOVA with Tukey post hoc test

chemokine expression, HEV may also alter leukocyte chemotaxis resulting in ineffective immune response and impaired viral clearance. In addition to elevated local inflammation, defective or inadequate placental development may explain HEV-1-associated pregnancy disorders. For instance, the HEV-1 associated decrease of CXCL-10 could impair the invasion/migration of the fetal trophoblast and remodeling of maternal spiral arteries, both of which are mandatory for successful placentation[46–49]. Our findings that conditioned media from infected samples induces tissue damage even in the absence of viral replication further highlight the involvement of the local environment in HEV-1 pathogenicity and underscore the importance of host factors that can also promote viral pathogenicity[17,45]. However, we cannot exclude the contribution of viral capsid protein and/or RNA in the induction and/or promotion of the release of soluble mediators. Nevertheless, the greater effects with HEV-1 clinical strain would suggest that both virus- and host-related factors contribute to the viral pathogenesis during pregnancy.

Despite the scarce information concerning HEV-3 during pregnancy, the lack of clinical manifestations suggests that the pro-inflammatory response to HEV-3 may correspond to a minor bystander effect that is harmless to fetal development. It's also possible that elements of the maternal immune system might limit HEV-3 replication at the maternal-fetal interface avoiding any pregnancy complications. As a matter of fact, we have previously demonstrated that Natural Killer cells from the maternal innate immune system can control the replication of human immunodeficiency virus and cytomegalovirus, preventing the development of congenital infection[29,50].

Most of pregnancy diseases related to HEV-1 infection have been reported mainly in the second and third trimester. Lessons learned from the TORCH pathogens have demonstrated that infections during early pregnancy are highly detrimental to fetal development. However, HEV-1-induced illness during the first trimester has not been well documented although it might be detrimental for pregnancy outcome. Even if our ex vivo model of first trimester pregnancy, similar to any other experimental model, has its limits, our findings within this model are the first suggesting that HEV-1 infection in early pregnancy may result in clinical symptoms that can be intermingled with pregnancy disorders, such as spontaneous abortion, or preeclampsia. The immunological and hormonal changes in later terms may then worsen the outcome of the pregnancy. Additional studies using samples from term pregnancy and/or in utero infected women as well as other viral strains warrant further investigations to fulfill our understanding of HEV pathogenicity.

In summary, our data suggest that the viral tropism and efficient replication combined with abnormal pro-inflammatory cytokine and chemokine secretion dictate the extent of tissue damage at the maternal-fetal interface and might be responsible for HEV-1-associated pregnancy disorders (Fig. 9). Beyond providing an explanatory mechanism for the severity of HEV-1 infection during pregnancy, we now present a new experimental model to study HEV infection, setting the stage for the testing and development of novel therapeutic strategies directed toward supporting human pregnancy.

## Methods

**Ethics statement**. This study was approved by the South-West & Outmer II ethical committee and was registered at the Ministry of Higher Education and Research (number DC-2016-2772). All participants provided prior written informed consent in agreement with the guidelines of the Declaration of Helsinki with experiments performed in accordance with approved guidelines.

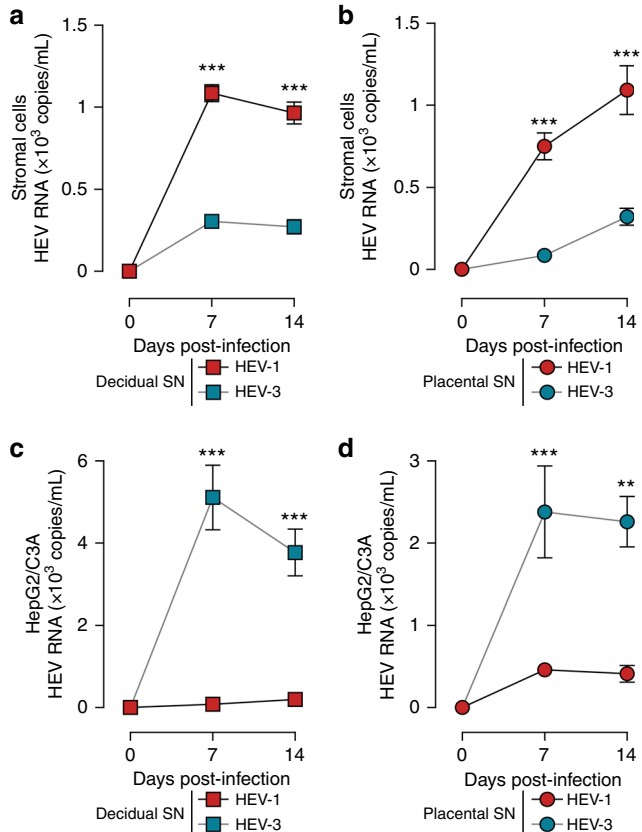

**Fig. 8** HEV-1 replication in explants derived from the decidua and placenta generates infectious progeny particles. Freshly isolated stromal cells **a**, **b**, or the HepG2/C3A cell line **c**, **d**, were challenged with culture supernatants (SN) from either HEV-1 (red), or HEV-3 (cyan) infected decidual **a**, **c**, or placental **b**, **d** explants. Viral replication was then followed over a 2-week period by RT-qPCR. Data represent means values ± S.E.M. of six independent donors. * denotes a statistical comparison made between HEV-1 and HEV-3 infected cells. **$P < 0.01$; ***$P < 0.001$ by two-way ANOVA with Bonferroni post hoc test.

**Tissue culture and isolation of primary cells**. First trimester pregnancy samples (8–12 weeks of pregnancy) were obtained from healthy women (aged 18–30 years) following an elective vaginal termination of pregnancy[51]. Briefly, 0.3 $cm^2$ of tissue explant was prepared from the *decidua basalis* (decidua) and placenta and maintained for up to 2 weeks in DMEM:F-12 supplemented with 10% fetal calf serum (FCS). Matched decidua and placenta samples were used in all the experiments.

For primary cell isolation, matrix digestion and cellular disaggregation were achieved with a 45 min (37 °C) digest with the enzyme, collagenase IV (Sigma-Aldrich, France). Mononuclear cells were then isolated from the cell suspension by Ficoll-Hypaque density gradient separation (Amersham Biotech) and plated overnight at 37 °C. Fibroblast-like stromal cells were enriched by successive rounds of mild trypsinization[29]. Human hepatocellular carcinoma HepG2/C3A cell line (ATCC CRL-10741) was maintained in DMEM culture medium supplemented with 10% FCS.

**HEV clinical strains and infection protocol**. HEV-1 and HEV-3 clinical strains were obtained at the acute phase of infection from the feces of a traveler returning from India and an autochthone infected patient, respectively. Patients were tested negative for HIV and Hepatitis A, B, and C. Fecal samples were diluted in DMEM and centrifuged at 1200×*g* for 10 min. Viral strains were recovered from the supernatant and passed through a 22 μm filter. HEV RNA was quantified by RT-qPCR with aliquots stored at −80 °C until use. Matched decidua and placenta explants were infected with the same HEV-1 or HEV-3 clinical strain.

Tissue were infected over a 24 h period (at 35.5 °C) with $2 × 10^7$ copies of HEV-1 or HEV-3 RNA (~MOI 10) in a 24-well plate and a final volume of 500 µl of DMEM:F-12 supplemented with 2% FCS; mock controls comprised uninfected explants. Each explant is then washed five times in PBS. The explants are then laid on collagen sponges in DMEM:F-12 supplemented with 10% FCS and cultured at 37 °C and 5% $CO_2$ humidified atmosphere[30].

Prior to infection, primary stromal cells isolated from either the decidua or placenta and HepG2/C3A cells were seeded in 6-well plates overnight ($5 × 10^5$ cells/well). Cells were then infected with $10^7$ RNA copies of either HEV-1, HEV-3 (~MOI 20), or were left uninfected in a final volume of 1 ml of DMEM:F-12 containing 2% FCS at 35.5 °C. After 24 h, the cell layer was rinsed five times with PBS and subsequently cultured in 2% FCS-medium at 35.5 °C.

At different time points post infection, half of the cell culture supernatant was collected and replaced with fresh media. Supernatants were stored at −80 °C for subsequent quantification of viral RNA and soluble mediators. For viral production kinetics, day 0 supernatant was collected after wash before culture. The residual value obtained for day 0 was subtracted from each kinetic point. Residual day 0 values were $1 × 10^5$ copies/ml and $3 × 10^5$ copies/ml for decidua and placenta respectively, regardless of the genotype.

**HEV quantification**. HEV RNA was extracted from culture supernatants using the QiaAmp viral RNA mini kit (Qiagen, Courtaboeuf, France). The RNA quantification was performed using one-step, real-time RT-PCR with a LightCycler 480 instrument (Roche Diagnostics, France). The following primers targeting the ORF2/ORF3 overlapping region were used: Forward primer (5′-GGTGGTTTCTGGGGTGAC-3′), Reverse primer (5′-AGGGGTTGGTTGGATG AA-3′) and the probe 5′–6-carboxyfluorescein (FAM)–TGATTCTCAGCCCTTCGC–6-carboxytetramethylrhodamine (TAMRA)–3′. The amplification efficiency was then calculated using the standard curve. RNA standards were designed using a conserved fragment within the ORF3 gene (70 nt) amplified from an HEV infected patient sample and cloned into pGEM.3Z vector. The fragment was then retro-transcribed using T7 RNA polymerase. The obtained positive strand was used as the RNA standard in all the quantitative RT-PCR experiments. A standard curve was generated from the serial tenfold dilutions of this RNA standard[52,53]. Data for normalized RT-qPCR values were presented as viral RNA copies/mL. The detection limit for this validated method is 100 HEV RNA copies/mL.

**Histological analyses**. After 5 days, mock, HEV-1, or HEV-3 infected decidual or placental tissue explants were fixed in 10% formalin, embedded in paraffin, and then 3-micron thick sections processed for histological analyses using hematoxylin and eosin (H&E) staining, in situ hybridization (ISH) and terminal deoxynucleotidyle transferase-mediated dUTP nick end-labeling (TUNEL) assay. Histology slides were scanned using the Panoramic 250 Flash system (3DHIS-TECH, Hungary). Images were processed using the Panoramic Viewer software (3DHISTECH, Hungary). The quantification of necrosis (H&E staining) in infected samples versus mock-infected controls was performed in ten high power fields per sample taken randomly.

For ISH of HEV RNA, tissue sections were processed according to the manufacturer's protocol (Advanced Cell Diagnostics, USA). The commercially available HEV-specific probes (V-HEV) were used to detect the positive strand RNA. RNase-free conditions were maintained during all steps. Slides were scanned using ×40 objective. For ISH quantification, the number of HEV-positive cells was counted in ten random regions that cover the whole slide and then reported to the area ($cm^2$) of each region.

The apoptotic cells were detected in situ by TUNEL assay using a Cell Death Detection Kit, TMR red (Sigma-Aldrich, France). The sections were processed according to the manufacturer's directions. The slides were further stained with 4,6-diamidino-2-phenylindole (DAPI, 1:5000 dilution, Sigma). Histology slides were scanned using ×20 objective. Apoptotic cells with red plot within the nuclei and normal cells with blue nuclei were identified in the processed sections. The quantification of apoptosis in infected samples versus mock-infected controls was performed in ten high power fields per sample taken randomly.

**Quantification of soluble factors**. Culture supernatants were collected 2 days post infection and stored at −80 °C for quantification of soluble mediators. Seventeen-multiplexed Affymetrix cytokine assays (Procarta/eBioscience, France) were used to quantify cytokines (IL-6, IL-15, IL-18, IFN-α, IFN-β, IFN-γ, TNF-α, and sICAM-1), chemokines (CCL-3, CCL-4, CCL-5, and CXCL-10), growth factors (G-CSF, GM-CSF, and VEGF-A), and metalloproteinases (MMP-2 and MMP-9) according to the manufacturer's protocol. Interferons assessment was conducted by Cytometric Bead Array (CBA) using the Human Type 1/2/3 Interferon Panel (5-plex) (Biolegend, USA) according to the manufacturer's protocol.

**Principal component analysis (PCA)**. PCA of IL-6, CCL-3, CCL-4, and CXCL-10 in both decidual and placental tissues were performed using ClustVis web tool (http://biit.cs.ut.ee/clustvis/). Values were centered, unit variance scaling was applied to rows and single value decomposition with imputation was used to calculate principal components.

**Preparation of conditioned medium (CM)**. Mock, HEV-1, or HEV-3 infected decidual or placental explant supernatants were collected 5 days post infection and then UV-irradiated for 30 min using Spectroline EF-140/F UV lamp (220 volts, 50 HZ, 17 Amps). The UV treatment was sufficient to abolish viral infection. Fresh

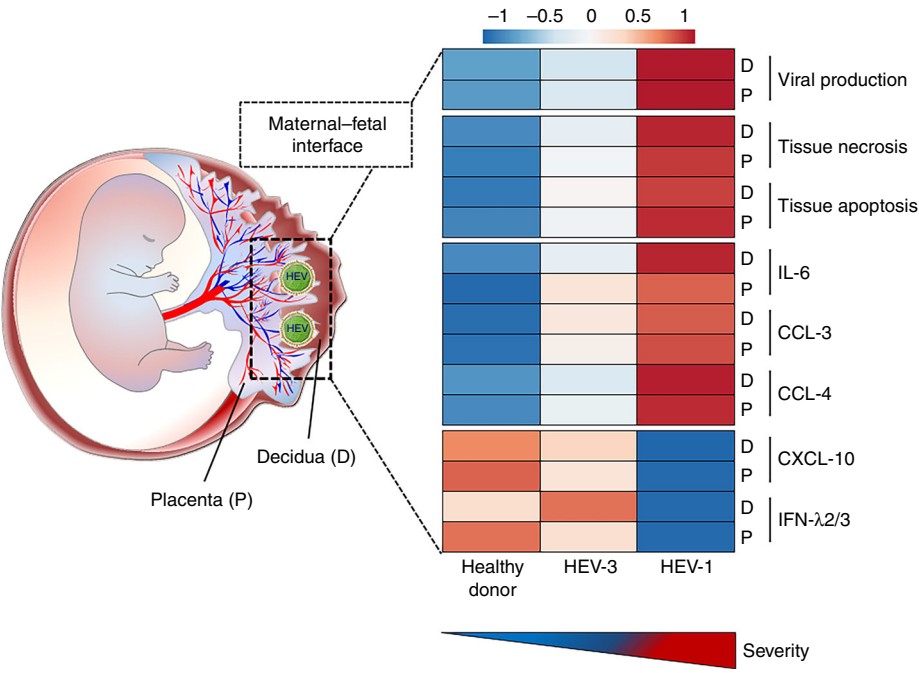

**Fig. 9** Genotype-specific pathogenicity of hepatitis E virus at the human maternal-fetal interface. Graphical abstract summarizing the differences observed between HEV-1 and HEV-3 infection in the maternal *decidua basalis* (D) and fetal placenta (P). Heatmap was generated using mean values for the different parameters analyzed in each study group

decidual and placental explants were then cultured for 5 days in decidual UV-CM and placental UV-CM, respectively, before further analyses.

**Immunocytochemistry and fluorescence microscopy**. Primary decidual and placental stromal cells or HepG2/C3A cells were plated on glass coverslips and then infected with either HEV-1, HEV-3, or were left uninfected. Seven days post infection, cells were fixed with 4% paraformaldehyde, permeabilized with 0.3% Triton X-100, and stained with antibodies against the HEV capsid protein (anti-ORF2, clones 1E6 and 4B2, 1:200 dilution, Clinisciences) and vimentin, a stromal cell marker (1:100 dilution, Cell Signaling Technology) for primary cells or α-tubulin for HepG2/C3A cells (1:250 dilution, Sigma-Aldrich). Bound primary antibodies were visualized by the addition of Alexa Fluor-conjugated, class-specific secondary antibodies (1:500 dilution, Invitrogen). Nuclei were stained with 4,6-diamidino-2-phenylindole (DAPI, 1:5000 dilution, Sigma-Aldrich). Z-stack images were acquired using an LSM710 confocal microscope (Carl Zeiss, Germany) with ×63 oil objective. Images were processed using the Imaris software (Bitplane AG, Switzerland).

**Identification of infectious progeny virions**. Culture supernatants were collected from HEV-1 or HEV-3 infected explants 5 days post infection with viral load quantified by RT-qPCR. Freshly isolated stromal cells, or the HepG2/C3A cell line (ATCC CRL-10741), were then infected with $5 \times 10^5$ RNA copies from each supernatant. After extensive washing, viral replication was followed over a 5-week period as described above.

**Ribavirin and interferon treatment**. Ribavirin (RBV) and recombinant IFN-λ1 and IFN-λ2 were purchased from Sigma-Aldrich and Peprotech France respectively. Tissue explants were treated with 50 μM of RBV or 100 ng/mL of IFN-λ after HEV infection and were maintained in tissue culture until the end point of the experiment.

**Statistical analyses**. All experiments were conducted on matched decidua and placenta samples from the same donor. The total number of independent donors for each experiment is indicated in the figure legend. Graphs represent mean values with error bars indicating the S.E.M. Statistical analyses were performed using the GraphPad Prism software 5 (GraphPad Software, La Jolla, USA). Two-way analysis of variance (ANOVA) with the Bonferroni post hoc test was used to compare the kinetics of HEV-1 and HEV-3 viral production in tissue explants, and in isolated primary cells from the same donors. Repeated measures ANOVA with the Tukey post hoc test was used to compare values for mock, HEV-1, and HEV-3 infected samples from the same donors. Correlations between soluble factor secretion and viral production were calculated with Spearman's rank correlation test. * denotes a statistical comparison made between HEV-1 and HEV-3 infected tissues or cells. # denotes a statistical comparison made between mock and HEV-1 or HEV-3

infected tissues or cells. *P*-values< 0.05 were considered to be significant (*/#*P* < 0.05; **/##*P* < 0.01; ***/###*P* < 0.001).

## Data availability
The authors declare that all the data supporting the findings of this study are available within the article and its supplementary information files, or are available upon reasonable request to the authors.

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

## Acknowledgements

The authors would like to thank MDs from the maternity hospitals, the histopathology core facility-UMS-US006 (T. Al-Saati and F. Capilla), and the CPTP cell imaging core facility (S. Allart). This work was supported by the INSERM, CNRS, and Toulouse III University (to N.J.-F., J.I.), and ANRS (ECTZ2844 to N.J.-F.). J.G. and Q.C. are supported by the ANRS, and the Chinese Scholarship Council PhD fellowships, respectively.

## Author contributions

J.G. designed the study and conducted experiments; Q.C., and J.S. conducted experiments. M.D. provided technical assistance; G.C. and C.L. provided the requisite clinical material for this study; J.I. and R.A.D. provided critical feedback; N.J.F. and H.E.C. jointly conceived and supervised this study, and wrote the manuscript. All authors approved the final version of the manuscript.

## Additional information

**Competing interests:** The authors declare no competing interests.

