## [Peer Review File · Nature Communications]

Reviewers' Comments:

Reviewer #1:

Remarks to the Author:

In the manuscript by Gouilly J et al., entitled "Genotype Specific Pathogenicity of Hepatitis E Virus at the Human Maternal-Fetal Interface", the authors described phenotypic differences in decidua and placenta explants after infection with HEV-1 or HEV-3. The authors described a new model of HEV infection using ex vivo culture of human tissues. The authors evaluated virus production and secretion of host factors such as cytokines, chemokines or metalloproteinases upon infection with HEV-1 or HEV-3. They observed changes in the secretion of some factors depending on the HEV genotype used for infection. In addition, they could efficiently infect stromal cells isolated from decidua or placenta with HEV-1 and to a lesser extent HEV-3.

Given the incidence of fatal outcome of the HEV-1 infection in pregnant women, the present study is highly relevant. Together, the use of explants model and HEV isolates from acutely infected patients is original. The manuscript is well-written and the figures well-designed. Of note, the discussion section is rather long as compared to the rest of the manuscript and may be streamlined.

Major issues:

- The authors infected the placenta or decidua explants either by HEV-1 or HEV-3 obtained from fecal samples of acutely infected patients. To my point of view, there are critical information missing in the manuscript which are essential to assess the quality of their study. The authors mentioned 6 donors in the figures legends however we don't know whether it concerns the HEV samples or the explants. Interpretation of the data may be different if it was one HEV isolate for each genotype that infected 6 explants or if it was 6 HEV isolates which served to infect one explant. Therefore, it is important to mention how many of each samples, including HEV-1 and HEV-3 as well as explants, have been used in the study.

- In addition to the RT-PCR data and to further characterize and validate their model, the authors may want to perform in situ hybridization to detect the viral RNA as recently described in two publications on mouse and human liver tissues, respectively references Allweiss L. et al. (J Hepatol. 2016;64:1033-1040.) and Lenggenhager D et al. (J Hepatol. 2017;67:471-479.). Additionally, such experiment may help to show if there is indeed a better tropism for HEV-1 in those tissues or if the replication itself is more important.

- The authors used mostly RT-PCR, which is extremely sensitive readout, to measure the released viral RNA in supernatant. In their results, the authors showed a "d0" value set at zero. The inoculum used in the assay is in the 10^7 range of infectivity and the values in supernatant at about 10^5 - 10^6 RNA copies/ml. Using RT-PCR it is likely that the RNA level measured late after infection are somehow "contaminated" by the input virus preparation. Is the sample taken prior or after inoculation with HEV? What does "extensive washing with PBS" (p15, l326) mean? In the same experimental settings, a control with the addition of an inhibitor of HEV replication (e.g. ribavirin) together with HEV-1 or HEV-3 infection would be appreciated to further validate the author's observations.

- The authors performed a cytokine secretion assay to evaluate the secretion of soluble factors in the different conditions summarized in Figure 2. In the Figure 3a, the authors performed correlation analysis for some of the factors that were differentially modulated by the two HEV genotypes between the amount of secreted factor and the viral RNA level in supernatant. From this analysis where they mixed together data obtained from HEV-1 and HEV-3 infections, the authors showed a correlation ("strong" as referred to authors p6, l124) for most of the factors which led them to conclude that the

altered-secretome is correlated to the viral production. By doing this analysis, the authors assumed that HEV-1 and HEV-3 have no differential effect on the secretion of these factors. However, as underlined by the authors and shown earlier in Figure 1, HEV-3 infection led consistently to a lower viral RNA level as compared to HEV-1 infection (see cloud of points for each genotype). Therefore, the analysis is biased. On one hand, the authors claimed that there are differences between the genotypes and in the other hand they were analyzing all data together taking into account only the HEV RNA level in supernatant. I would advise the authors to analyze separately HEV-1 and HEV-3 data for the correlation analysis.

Minor issues:

- The authors used fecal samples as inoculum for HEV infection. However, it is known that HEV in feces is present as naked virion while the virus in blood circulates as a "quasi-enveloped" particle. Did the authors try to infect their model with HEV retrieved from serum samples? As the source of virus for the infection of placenta or decidua in vivo shall be originate from blood, it would also be important to address this question to validate the infection model.
- The authors performed a cytokine secretion assay to evaluate the secretion of soluble factors in the different conditions. The data are summarized in Figure 2 by histograms and are further shown, for a selection of them, in Figure 3B as a heatmap. These are redundant information that are probably not necessary.
- p3, l43: "(HEV)" does not referred to "Hepatitis E" but to "Hepatitis E virus"
- p3, l49: "(15 to 30% of cases)" is referring to fatal outcome (FHF) and not to HEV-1 infection during pregnancy, therefore the authors may want to move it after "FHF" in the text.
- p8, l156: The authors may want to replaced "infectivity" by "infection".

Reviewer #2:

Remarks to the Author:

The paper of Gouilly et al describes the 2- to 4- fold higher replication of a HEV gt1 clinical strain compared to a HEV gt3 clinical strain in human decidua and placenta explant tissues. These marginal (less than 1-log) higher HEV gt 1 viral loads are also observed in fibroblast stroma cells isolated from the same tissues. HEV replication differences are associated with changes in the secretome in culture media and seem to lead to more histological tissue damage in placental explant tissues, without cytopathic effects in fibroblast stroma cells. Overall, the observation of replication of HEV gt1 in placental tissue is not novel, as also mentioned by the authors. No mechanistic insight is provided into the reasons of the apparent higher replication efficiency, neither are there any efforts to characterize the pathogenesis in more depth. The paper does therefore not significantly advance our understanding of the peculiar clinical differences between HEV gt1 and gt3 strains.

Major points:

1/ Corroboration of the observations in clinical material and in vivo models is essential. Higher HEV gt1 replication compared to HEV gt3 in human hepatocytes in chimeric mice has been demonstrated previously. Rhesus monkeys are susceptible to both HEV gt3 and HEVgt1 infection and would be a suitable model to examine in vivo replication in placenta tissue. Is it possible to examine placenta's of

HEV gt3 infected women in the Southern part of France? I assume some of the pregnant women in a high incidence region would be infected with HEV gt3. As the authors mention, Bose et al (Journal of General Virology (2014), 95, 1266) have demonstrated extrahepatic replication of HEV gt1 in ex vivo placenta tissue. This paper confirms that HEV gt1 can be propagated in placental tissue in vitro and adds only data on cytokine/chemokine derangements.

2/ Several cytokines and chemokines are differentially induced and the authors propose that these contribute to the viral pathogenesis observed. They do not dissect out which pathway or cytokine is responsible for the observed higher necrosis nor the cellular mediators producing these cytokines. Are immune cells involved? A first option could be to deplete or block some of the cytokines/chemokines that might cause these effects and see whether they can impact the observed necrosis? Another option could be to add corticosteroids to see whether the effects on the placental tissue can be reversed. These insights are essential for the current management of HEV gt1 infected pregnant women.

3/ Virological differences are rather small, 2- to 4-fold differences at most. Generally spoken clinical relevant differences are considered to be 1 log or more. Can the authors explain why they consider these small differences as relevant for the clinical differences? Replication differences are more pronounced up to day 7 after inoculation, but start to diminish thereafter. In vitro culture of faeces derived HEV on different hepatoma or adenocarcinoma cell lines generally follows a different pattern, with steady increase during the first 4 weeks and a plateau phase thereafter. Given the claim of a new in vitro model for HEV gt1 propagation, the authors should more extensively characterize the stability of the system: How long can these cultures be maintained? Can the viruses be passaged? At least extension up to 4 weeks is necessary to prove that the initial different kinetics are not lost during prolonged culture as seem to be the case at d12 in Fig 1b and d14 in Fig6a/b.

4/ The observed pathological effects in placental tissue are the cornerstone of the paper. However, it is only evaluated by standard histology, which is subjective and only semi-quantitative. Evidence of necrosis is provided in Fig 1c, but only one denuded syncytiotrophoblast layer is shown, with limited other signs of necrosis. More extensive characterization of the pathogenesis is needed by e.g. different cell death or cell stress assays (LDH-release, single cell viability evaluation by FACS, ATP-release...). In addition, no direct cytopathic effects are seen in isolated fibroblast stromal cells. How do the authors explain these differences? What is the mechanism causing the placental necrosis. Is it infiltration by immune cells. Again more profound examination of the pathophysiological mechanism should be performed.

Minor points:

1/ Number of infected cells between HEV gt1 and HEV gt 3 inoculated placenta. Kinetics of spread? Are there differences?

2/ Which cells produce the different cytokines? Are the infected cells secreting these cytokines? Are there still immune cells included in the placental tissue?

3/ There were no type I and II IFN detected in culture media after infection. There is an ongoing debate about the induction of ISG by HEV gt3 and gt1 in infected cells. Some authors mention induction in vitro in cell-lines, others do not see innate immune responses in human hepatocytes. Can the authors discuss their findings? Are the cells not sensing the infection? How do they explain the changes in the secretome? What about type III IFNs which have recently been shown to be increased in hepatoma cells (Yin et al. PLOS Pathogens 2017).

4/ There seem to be differences in the inocula used for the initial propagation (7 log IU) and the infectivity assay of progeny virions (5 x 10E5 IU). Why?

5/ Different scales are used in the bar graphs of decidua and placenta (eg Fig 2a/b), which makes comparison difficult, for the same cytokine an identical scale should be used e.g. IL-6.

Reviewer #3:

Remarks to the Author:

General comments

The authors have studied the important question why HEV genotype1 is so pathogenic during pregnancy while the closely related, less pathogenic HEV genotype 3 does not affect pregnant women and the fetus. HEV is difficult to grow in cell cultures. The authors chose to inoculate explants of human placenta and decidua with HEV-1 or -3 samples from patient feces to generate stock virus. It is not clear why they did not try to grow the patient-derived HEV-1 and HEV-3 strains first in cell lines optimized for HEV cultivation like HepG2/C3A or the lung carcinoma line A549. Importantly, they show that increasing HEV RNA titers appear in the supernatants of the inoculated explants suggesting susceptibility of these tissues for both HEV-1 and -3. The main point is that HEV-1 seems to grow better than HEV-3 in these tissues, which could explain why HEV-1 is more pathogenic for pregnant women and the fetus. The question remains whether the rather moderate differences in the growth rate by a factor of 3-4 between HEV-1 and -3 is sufficient to explain the drastic differences in real patients. The proof that HEV really grows in these tissues is incomplete (point 1b) because the titer of progeny virus is much lower than the amount of input virus. Thus, additional evidence for replication is needed. The extensive testing of cytokines and other cellular factors changed by the addition of HEV is interesting but insufficient to corroborate conclusions concerning HEV replication.

Specific points

Major points

1. Fig 1a, b.

a. What was the multiplicity of infection, i. e. the copy number per cell number? It is not clear to how many cells and in which volume the inoculum was given.

b. From the figures one could guess that the maximum amount of the newly formed virus was ca. 8 times less than the input virus. Thus, it cannot be excluded that the virus detected was transiently bound to the cells and then released without real replication. Detection of minus strand RNA or of subgenomic RNA would be a better proof of replication.

c. Does day 0 p. i. mean the time immediately after inoculation of 2×10^7 /mL HEV RNA copies and thorough washing? Was there really no HEV RNA detectable in view of a detection limit of 100/mL?

c. Differences in histopathology occurred mostly in syncytiotrophoblasts (HEV-1) and secondarily in villus cores (HEV-3) from 6 donor tissues at 5 days post-infection. Broken syncytiotrophoblasts can result from handling and is atypical of viral infection. How many breaks were observed per explant and was there an increase over time? For HEV-3, by what mechanism did necrotic zones appear in villus cores? What does fold-change mean in pathology of infected tissues relative to controls? Blebbing of syncytiotrophoblasts occurred in controls, HEV-1 and -3. What is the difference?

2. L93. "privileged replication sites" means that there are other sites of replication, but in this part of the text no other sites are mentioned. Since the tissue tropism is the major point of the paper, the authors should re-organize the text to show another tissue or cell culture for comparison as they did later.

3. Fig. 3.

a. The legend should mention at which time point of infection the various cultures were analysed.

b. The panel for IL-6 does not suggest that the virus production has an influence on IL-6 secretion even if the p-value is 0.04. See also lines 124, 125.

c. L134-141. The UV-treated conditioned medium still contains HEV RNA which by itself may induce cytokines. Can this effect be excluded?

4. Fig. 4b. What was the multiplicity of infection for the stromal cells? See point 1b.

5. Fig. 4c.

a. The number of HEV capsid producing stromal cells seems to be low. A quantitative counting of positive cells per total cell number would be desirable.

b. Is this staining pattern really typical for an HEV infection?

c. A convincing positive and negative control is missing, e.g. infected and uninfected HepG2/C3A cells.

d. The appearance of decidual and placental stromal cells is similar. Usually decidual stromal cells are cultured under conditions that reflect the uterine origin and could effect virus production, which was not done here, but would be relevant to infection HEV during pregnancy.

6. Fig. 6

a. What was the multiplicity of infection?

b. Testing the infectivity in new decidual or placental explants would have been more convincing to prove the differential infectivity.

7. Methods. The calibration of the qPCR for HEV RNA is not sufficiently described. How were the internal standards for HEV RNA generated and calibrated? How does the number of HEV RNA molecules relate to HEV particles and infectious virions?

Minor points

d. Spell out abbreviations CPTP, and UCSF, CHU, IFB in the affiliations.

e. L31 and later. Replace "non-pathogenic" by "less pathogenic". HEV-3 is pathogenic for ca. 0.1 % of the infected subjects causing symptomatic acute or chronic hepatitis.

f. L44. A recent reference to the current taxonomy of the Hepeviridae would be useful.

g. L82. The gestational age of the explants should be mentioned because they change considerably during pregnancy.

h. L83. It ought to be mentioned here and not only in the methods that the virus came from feces of patients with symptomatic (?) hepatitis E.

i. Fig. 6 Typo: placental

Point-by-point response to the reviewer comments.

Reviewer 1:

Summary of comments from this reviewer: The reviewer greatly appreciated our experimental design and found our study highly relevant in the context of the fatal outcome of the HEV-1 infection in pregnant women. He/she appreciated the fact that we used a clinically relevant model to address the infection of placenta or decidua explants by HEV-1 or HEV-3 obtained from acutely infected patients. However, he/she raised concerns regarding some critical information that was missing in the manuscript.

Response to Reviewer's general comments:

Of note, the discussion section is rather long as compared to the rest of the manuscript and may be streamlined.

Response:

We followed the reviewer suggestion and have shorten the discussion accordingly.

Major points:

The authors mentioned 6 donors in the figures legends however we don't know whether it concerns the HEV samples or the explants. Interpretation of the data may be different if it was one HEV isolate for each genotype that infected 6 explants or if it was 6 HEV isolates which served to infect one explant. Therefore, it is important to mention how many of each samples, including HEV-1 and HEV-3 as well as explants, have been used in the study.

Response:

We thank the reviewer for alerting us that the text concerning the number of samples might be misleading. Imported HEV-1 and domestic HEV-3 strains for inoculation were recovered from the feces of an acutely HEV-1-infected traveler returning from India and an acutely autochthonous HEV-3-infected patient. The same cleansed fecal suspension was used throughout the study. In this study, we used matched decidual and placental explants from independent pregnancy terminations. We have fully revised the Methods section and figure legends to include details concerning the number of matched samples for decidua and placenta as well as details about the clinical strains of HEV-1 and HEV-3. This information is incorporated in the p17-18 for M&M.

*In addition to the RT-PCR data and to further characterize and validate their model, the authors may want to perform *in situ* hybridization to detect the viral RNA as recently described in two publications on mouse and human liver tissues, respectively references Allweiss L. et al. (J Hepatol. 2016;64:1033-1040.) and Lenggenhager D et al. (J Hepatol. 2017;67:471-479.). Additionally, such experiments may help to show if there is indeed a better tropism for HEV-1 in those tissues or if the replication itself is more important.*

Response:

We have followed the reviewer's suggestion and performed *in situ* hybridization to detect the viral RNA in matched decidua and placenta tissue samples, using a set of probes that covers the whole HEV genome. These experiments clearly revealed that HEV-1 replicates much better than HEV-3 in the maternal-fetal interface, be it *decidua basalis* or placenta, complementing and further supporting the notion of better tropism for HEV-1 in both tissues shown originally by RT-PCR. These results are now reported in lines 98-107 of the revised manuscript and Fig. 1c,d.

The authors used mostly RT-PCR, which is extremely sensitive readout, to measure the released viral RNA in supernatant. In their results, the authors showed a "d0" value set at zero. The inoculum used in the assay is in the 10⁷ range of infectivity and the values in supernatant at about 10⁵-10⁶ RNA copies/ml. Using RT-PCR it is likely that the RNA level measured late after infection are somehow "contaminated" by the input virus preparation. Is the sample taken prior or after inoculation with HEV ? What does "extensive washing with

PBS” (p15, l326) mean ? In the same experimental settings, a control with the addition of an inhibitor of HEV replication (e.g. ribavirin) together with HEV-1 or HEV-3 infection would be appreciated to further validate the author’s observations.

Response:

Tissue contact with the inoculum is carried out for 24h as stated in M&M. As for extensive washing, explants are transferred to a new 6 well plate and washed 5 times in 5ml PBS with five changes to new plastic plates before culture. The explants are then laid on top of collagen sponges¹ in the presence of 1ml of complete culture medium. Supernatant “day 0” is collected after infection and washing but before culturing. This residual value obtained at the so-called “day 0” is in the range of 10⁵ copies/ml (1x10⁵ for decidua and 3x10⁵ for placenta). Since these basal values do not change the observed differences between HEV-1 and HEV-3 replication kinetics, we subtracted them from each time point in order to have the most accurate and just representation of viral replication. However, if the reviewer feels that this should be shown, we could modify the graph in Fig. 1 accordingly and show the baseline or add it as a supplementary figure. These technical details are now included in the M&M section (p18-19).

At different time points post-infection, half of the cell culture supernatant was harvested and replaced with fresh media. Under these conditions, we observed the increase of viral RNA within culture supernatant that highlights an active viral replication rather than passive release over time or contamination by the input virus preparation.

These technical clarifications are now included in M&M section (p18-19). To further confirm this notion, we performed additional controls of viral replication using ribavirin as suggested by the reviewer. The new data included in Supplementary Fig. 1 and lines 91-97 of the revised manuscript show a significant inhibition of viral replication in the presence of 50µM RBV. Taken together, these results, in addition to the RNA scope experiments, provide solid evidence of active viral replication of HEV genotype 1 and 3 replication within decidual and placental tissue.

- The authors performed a cytokine secretion assay to evaluate the secretion of soluble factors in the different conditions summarized in Figure 2. In the Figure 3a, the authors performed correlation analysis for some of the factors that were differentially modulated by the two HEV genotypes between the amount of secreted factor and the viral RNA level in supernatant. From this analysis where they mixed together data obtained from HEV-1 and HEV-3 infections, the authors showed a correlation (“strong” as referred to authors p6, l124) for most of the factors, which led them to conclude that the altered-secretome is correlated to the viral production. By doing this analysis, the authors assumed that HEV-1 and HEV-3 have no differential effect on the secretion of these factors. However, as underlined by the authors and shown earlier in Figure 1, HEV-3 infection led consistently to a lower viral RNA level as compared to HEV-1 infection (see cloud of points for each genotype).

Therefore, the analysis is biased. On one hand, the authors claimed that there are differences between the genotypes and in the other hand they were analyzing all data together taking into account only the HEV RNA level in supernatant. I would advice the authors to analyze separately HEV-1 and HEV-3 data for the correlation analysis.

Response:

We agree with the reviewer that an unbiased analysis would be to run separate correlation studies. We now analysed HEV-1 and HEV-3 data separately. These new analyses confirmed our original findings and are now shown in Fig. 3c,d for HEV-1 and in Supplementary Fig. 2 for HEV-3. The results are described in the text (lines 153-161).

Minor points:

- The authors used fecal samples as inoculum for HEV infection. However, it is known that HEV in feces is present as naked virion while the virus in blood circulates as a “quasi-enveloped” particle. Did the authors try to infect their model with HEV retrieved from serum samples? As the source of virus for the infection of placenta or decidua in vivo shall be originate from blood, it would also be important to address this question to validate the infection model.

Response:

Although the mechanisms with which HEV infects target cells are not clearly understood, we agree with the reviewer that *in vivo* infection of the decidua and placenta is very likely from quasi-enveloped virions within the maternal blood circulation. However, previous studies clearly showed that serum-derived virions are much harder to propagate *in vitro* and *in vivo* and it has been revealed that the establishment of HEV infection in tissues occurs through stool-derived HEV-1 but not through serum-derived virus². Furthermore, our work and data from Vanwollehem group^{3,4} reveal that the infection with feces-derived HEV is much higher than that of plasma-derived virus. Therefore, to demonstrate the preferential tropism of HEV-1 over HEV-3 in placental tissues and bypass technical hurdles, we used a cleansed fecal-derived virus to infect primary tissues and cells from the maternal-fetal interface.

- The authors performed a cytokine secretion assay to evaluate the secretion of soluble factors in the different conditions. The data are summarized in Figure 2 by histograms and are further shown, for a selection of them, in Figure 3B as a heatmap. These are redundant information that are probably not necessary.

Response:

We agree with the reviewer that the Heatmap does not yield more information and have removed it accordingly.

- p3, 143: “(HEV)” does not referred to “Hepatitis E” but to “Hepatitis E virus”

Response:

This issue is now corrected in the revised manuscript.

- p3, 149: “(15 to 30% of cases)” is referring to fatal outcome (FHF) and not to HEV-1 infection during pregnancy, therefore the authors may want to move it after “FHF” in the text.

Response:

This has been corrected in the revised manuscript.

- p8, 1156: The authors may want to replaced “infectivity” by “infection”.

Response:

The term infectivity has been replaced by infection in P8 of the revised manuscript.

Reviewer 2:

Summary of comments from this reviewer: We thank the reviewer for his/her critical review and comments on our study. We believe that a more extensive knowledge of the pathogenesis of HEV-1 during pregnancy would be valuable to the management and healthcare. The additional experiments that we have performed now significantly improve the quality of the manuscript and our understanding of the differential pathogenesis of HEV-1 and HEV-3 at the maternal-fetal interface.

Response to Reviewer’s general comments:

The paper of Gouilly et al describes the 2- to 4- fold higher replication of a HEV gt1 clinical strain compared to a HEV gt3 clinical strain in human decidua and placenta explant tissues. These marginal (less than 1-log) higher HEV gt 1 viral loads are also observed in fibroblast stroma cells isolated from the same tissues. HEV replication differences are associated with changes in the secretome in culture media and seem to lead to more histological tissue damage in placental explant tissues, without cytopathic effects in fibroblast stroma cells. Overall, the observation of replication of HEV gt1 in placental tissue is not novel, as also mentioned by the authors. No mechanistic insight is provided into the reasons of the apparent higher replication efficiency, neither are there any efforts to characterize the pathogenesis in more depth. The paper does therefore not significantly advance our understanding of the peculiar clinical differences between HEV gt1 and gt3 strains.

Response:

Currently most of our knowledge about the pathogenesis of HEV during pregnancy come either from case reports or longitudinal studies using peripheral blood samples. Given the difficulty to propagate HEV, most of published data have used cell lines, cell culture-adapted viral genomes and transfection experiments. Whilst the data obtained by these approach have proven valuable to characterize several aspects of HEV infection, their overall sequence included multiple mutations/deletions scattered throughout the genome.

A unique study from Bose et al.⁵ has identified viral components in placental tissue from HEV-1-infected women, yet differential replication and pathogenesis of HEV genotypes was beyond the scope of that study.

The novelty of our study resides in the following points:

- i) the use of clinical strains that have not been propagated *in vitro*.
- ii) the use of a clinically relevant *ex vivo* model of matched decidua and placenta tissues as well as primary cells from the same donors.
- iii) first study to address the differential replication and pathogenesis of HEV genotypes at the maternal-fetal interface.
- iv) first evidence that HEV-1 and HEV-3 replicates within the maternal *decidua basalis* and HEV-3 within the fetal placenta.
- v) the characterization of local secretion upon HEV infection since peripheral responses do not accurately reflect the events occurring at the maternal-fetal interface.
- vi) both viral factors and changes in the cytokine microenvironment contribute a great deal to the pathogenesis of the HEV-1 during pregnancy.

Besides providing the first evidence that HEV-1 has a higher tropism than HEV-3 for the maternal-fetal interface, our original findings suggest that in addition to the blood circulation, the maternal *decidua basalis* might serve as a replication platform before the dissemination of the virus to the fetal compartments. This notion is further strengthened by the transfer experiments showing that the efficiency and tropism of the initial strains for the maternal-fetal interface are preserved in the newly synthesized HEV-1 virions.

To further validate the viral replication in our model, we conducted as suggested by the reviewers, ribavirin inhibition experiments and *in situ* hybridization of viral RNA in both decidua and placental explants.

Moreover, we added TUNEL experiments to better characterize the tissue induced injury.

In an effort to characterize the pathogenesis in depth, we provide new experiments showing that the genotype-specific discrepancies between HEV-1 and HEV-3, could be related to differences in type III interferon response. These new data have been included in the revised manuscript as Fig. 5 and described in the revised manuscript (Lines 181-204).

Altogether, we strongly believe that our findings within the context of current state-of-the-art do provide new insights into the differential HEV pathogenesis at the maternal-fetal interface.

Major points:

1/ Corroboration of the observations in clinical material and in vivo models is essential. Higher HEV gt1 replication compared to HEV gt3 in human hepatocytes in chimeric mice has been demonstrated previously. Rhesus monkeys are susceptible to both HEV gt3 and HEVgt1 infection and would be a suitable model to examine in vivo replication in placenta tissue. Is it possible to examine placenta's of HEV gt3 infected women in the Southern part of France? I assume some of the pregnant women in a high incidence region would be infected with HEV gt3. As the authors mention, Bose et al (Journal of General Virology (2014), 95, 1266) have demonstrated extrahepatic replication of HEV gt1 in ex vivo placenta tissue. This papers confirms that HEV gt1 can be propagated in placental tissue in vitro and adds only data on cytokine/chemokine derangements.

Response:

Animal models have proven useful to study several aspects of the HEV infection, including cross-species

infection, tissular tropism, liver injury and mechanisms of virus replication as well as vaccine development. Unfortunately, they didn't allow to understand the effect of HEV during pregnancy and how it may cross the placental barrier. Human pregnancy differs from mouse pregnancy because of the unique fetal trophoblast invasion into the maternal *decidua basalis* as well as an increasing number of factors that are specific to human placentation^{6,7}. While higher primates could be used as surrogate models for human pregnancy, reports from Purcell et al. did neither reproduce HEV-induced disease in pregnant dams nor did it hamper fetal development or survival⁸. Moreover, high costs and ethical restraints make these models of very limited access. Therefore, many aspects of human placentation can only be addressed using human cells and tissues.

We cannot exclude the fact that pregnant women in the Southern part of France could be infected with HEV-3. However, to date there have been no case reports on pathological pregnancy due to HEV-3 infection in Europe including the southern part of France. The *decidua basalis* and placenta samples were obtained from uninfected healthy women undergoing elective pregnancy termination. For ethical considerations, even unrelated pathological pregnancies were excluded from the study.

2/ Several cytokines and chemokines are differentially induced and the authors propose that these contribute to the viral pathogenesis observed. They do not dissect out which pathway or cytokine is responsible for the observed higher necrosis nor the cellular mediators producing these cytokines. Are immune cells involved? A first option could be to deplete or block some of the cytokines/chemokines that might cause these effects and see whether they can impact they can impact the observed necrosis. Another option could be to add corticosteroid to see whether these effects on the placental tissue can be reversed. These insights are essential for the current management of HEV gt1 infected pregnant women.

Response:

In the present study, we show that the HEV-1 infection modulates the secretion of several soluble mediators including the cytokines and chemokines. Besides the originally tested cytokines, chemokines and growth factors, we added in the revised manuscript the quantification of type III interferons given their role in protecting the placenta from viral infections. We'd like to point out the central role of these soluble mediators during pregnancy. Our choice of the cytokine/chemokine profile tested is motivated by the lessons learned from other viral infections and/or pregnancy-associated pathologies. Different cells at the maternal-fetal interface including immune and non-immune cells produce the soluble mediators. It is quite well established through several studies including ours that these factors act in a synergistic manner to promote the placental development and fetal growth. The impairment of this cytokine balance can result in dysfunctions of the maternal-fetal interface^{1,9-15}. We are aware that depletion and add-back experiments can be an easier way to provide straight answers, which are common assays outside the pregnancy field but these kind of experiments are not adequate in the context of the maternal-fetal interface, as they will induce severe damage by themselves. Therefore, we ascertained that the HEV-induced changes to the cytokine microenvironment impair the maternal-fetal interface by analyzing the effects of the UV-irradiated supernatants (conditioned media, CM) on new samples of the maternal-fetal interface. We cannot exclude the sensing of viral proteins and/or RNA contained in the UV-treated conditioned media that might act as a feedback loop amplifying the changes in the local cytokine balance and further contributing to the observed tissue damage. Altogether, our findings show that soluble mediators within the CM can contribute to tissue injury independently of active viral replication. These points are now discussed in the revised version (Lines 163-176 and 315-322).

We agree with the reviewer that assessing the effects of corticosteroid might be interesting in the context of providing insights for the current management protocols but our study was rather designed to provide insights into the differential pathogenesis of HEV-1 and HEV-3. Therefore, in addition to the reasons stated above, we believe that these assays despite their potential interest are outside the scope of the present study.

3/ Virological differences are rather small, 2- to 4-fold differences at most. Generally spoken clinical relevant differences are considered to be 1 log or more. Can the authors explain why they consider these small differences as relevant for the clinical differences? Replication differences are more pronounced upto day 7 after inoculation, but start to diminish thereafter. In vitro culture of faeces derived HEV on different hepatoma or adenocarcinoma celllines generally follows a different pattern, with steady increase during the

first 4 weeks and a plateau phase thereafter. Given the claim of a new in vitro model for HEV gt1 propagation, the authors should more extensively characterize the stability of the system: How long can these cultures be maintained? Can the viruses be passaged? At least extension upto 4 weeks is necessary to proof that the initial different kinetics are not lost during prolonged culture as seem to be the case at d12 in Fig 1b and d14 in Fig6a/b.

Response:

We agree that this by itself might not be sufficient to explain the drastically different outcome in pregnant women, but this difference was clearly associated with prominent tissue damage in HEV-1 infected samples. Furthermore, viral replication was correlated to impairment of the cytokine balance at the maternal-fetal interface. The disruption of the cytokine microenvironment has been also reported in pregnancy complications such as miscarriage, pre-eclampsia and IUGR¹¹⁻¹⁵. This notion is further supported by experiments in Fig. 4 using UV-inactivated conditioned media. Independently from viral replication, the induction of tissue damage by CMV - albeit to a lesser extent than replicative virus - is much higher with HEV-1 than with HEV-3. Interestingly, previous reports have suggested that active HEV replication in placenta in conjunction with other factors contribute to the severity of the disease during pregnancy⁵. Even a moderate difference in the viral growth rate might result in a HEV-1-associated severe pregnancy outcome. As a whole, our data suggest that the advantageous replication of HEV-1 associated with the cytokine storm would work in concert to inflict such a drastic outcome of the pregnancy. These clarifications are included in the revised manuscript (lines 315-322).

We agree with the reviewer that in the tissue explants the differences of replication are more pronounced up to day 7. They then either plateaued or start to decline for the tissues and continue to increase for up to 14 days for primary cells of the maternal-fetal interface. All our experiments were carried out with matched maternal/fetal tissues and primary isolated cells from the first trimester of pregnancy. By contrast to the cell line cultures that can be maintained for 50-60 days, the maximum culture time is two weeks for primary cells and 8-10 days for tissue explants. Our clinically relevant model of maternal-fetal interface is very reliable and has been used successfully to investigate how infectious pathogens (CMV, ZIKV, HIV, Listeria and Plasmodium) disseminate to the placental barrier. Regarding HEV, the ribavirin inhibition experiments (Supplementary Fig. 1) along the *in situ* hybridization (Fig. 1c, d) and the thorough characterization of the HEV-1-induced tissue injury (Fig. 2) provided the proof of concept that our model is highly valuable.

4/ The observed pathological effects in placental tissue are the cornerstone of the paper. However, it is only evaluated by standard histology, which is subjective and only semi-quantitative. Evidence of necrosis is provided in Fig 1c, but only one denuded syncytiotrophoblast layer is shown, with limited other signs of necrosis. More extensive characterization of the pathogenesis is needed by e.g. different cell death or cell stress assays (LDH-release, single cell viability evaluation by FACS, ATP-release...). In addition, no direct cytopathic effects are seen in isolated fibroblast stromal cells. How do the authors explain these differences? What is the mechanism causing the placental necrosis. Is it infiltration by immune cells. Again more profound examination of the pathophysiological mechanism should be performed.

Response:

We are grateful to the reviewer for raising this point. We have now conducted TUNEL experiments to provide solid proof of tissue injuries following HEV infection. The HEV *per se* is not cytopathic and our data on stromal cell and cell line culture are in agreement with this notion^{16,17}. However tissue injury upon HEV-1 and HEV-3 infection were previously described and incriminated soluble factors and immune cells^{16,18-20}. In our experiments, isolated stromal cells and tissue cultures did not show the same secretion profile. This difference may be responsible of the cytopathic effect observed in tissue culture. However, the viral induced mitochondrial damage and activation of the caspase family member's reported by Soomro et al.²¹ might very well be involved in the observed apoptosis and necrosis at the maternal-fetal interface. Furthermore, we cannot exclude that hyper-activation of immune cells mainly maternal decidual Natural Killer (dNK) cells might be responsible for the tissue injury *in vivo*. In fact, we have previously reported, in the context of human cytomegalovirus infection, that the dNK cells become killer cells, migrate into the infected fetal placenta and induce lysis of the infected cells²². Although not applicable in placenta explant culture model, this mechanism might occur in decidua explants.

Minor points:

1/Number of infected cells between HEV gt1 and HEV gt 3 inoculated placenta. Kinetics of spread? Are there differences?

Response:

The number of the infected cells has been evaluated for each viral strain using ISH experiments. The ISH data confirms the significantly higher infection with HEV-1 inoculum than HEV-3 (Fig. 1c,d and lines 98-107).

2/ Which cells produce the different cytokines? Are the infected cells secreting these cytokines? Are there still immune cells included in the placental tissue?

Response:

As shown in the new Figure 6 isolated stromal cells of the *decidua basalis* and the placenta secrete the IL-6, sICAM, CCL-3, CCL-4, G-CSF, VGF-A and MMP-2. However, we cannot exclude that the maternal immune cells such as the decidual NK cells and Macrophages or fetal Hofbauer cells also produce some of these soluble mediators^{1,22,23}.

3/ There were no type I and II IFN detected in culture media after infection. There is an ongoing debate about the induction of ISG by HEV gt3 and gt1 in infected cells. Some authors mention induction in vitro in cell-lines, others do not see innate immune responses in human hepatocytes. Can the authors discuss their findings? Are the cells not sensing the infection? How do they explain the changes in the secretome? What about type III IFNs which have recently been shown to be increased in hepatoma cells (Yin et al. PLOS Pathogens 2017).

Response:

We thank the reviewer for bringing this up. We have conducted additional experiments to quantify type I, II and III IFNs produced by the decidual and placental explants (Fig. 5, Lines 181-204). We show that HEV-1 and HEV-3 infections does not affect the production of IFN- α 2, IFN- β or IFN- γ . These findings are in agreement with previous reports showing that HEV-1 or HEV-3 infection did not impair type I or type II IFNs in infected human liver chimeric mice²⁴. However, HEV-1 infection significantly impaired the production of IFN- λ 1 and IFN- λ 2/3 in the decidual explants and IFN- λ 2/3 in the placental explants. Contradictory results have been reported about the link between HEV infection and the induction of IFNs and ISGs according to the experimental model and the used viral strain²⁴⁻²⁸. The link between HEV infection and IFNs is not fully understood at the maternal-fetal interface. Herein, the significant decrease in type III IFN production was associated with a decreased level of the IFN-induced protein (CXCL10). Furthermore, HEV-1 infection of the decidual explants was inhibited by the presence of IFN- λ 1 or IFN- λ 2, whilst the infection of the fetal placenta was decreased by IFN- λ 2 treatment. Taken together our findings suggest that HEV-1 impairs the type III IFNs and CXCL-10 to replicate at the maternal-fetal interface. The type III interferons have emerged as a defense mechanism against several TORCH pathogens that can reach the maternal-fetal interface²⁹, their down-modulation by HEV-1 infection might constitute a mechanism of immune escape. Taken together, our data highlight the type III IFN as a potential defense mechanism against HEV-1 infection that can be exploited as therapeutic agent to prevent the disastrous outcome of pregnancy.

4/ There seem to be differences in the inocula used for the initial propagation (7 log IU) and the infectivity assay of progeny virions (5 x 10E5 IU). Why?

Response:

We agree with the reviewer that the inoculation in the transfer experiments using decidual and placental progeny virions was performed with 100-fold lower amount of viruses compared to the original infection. This is inherent to the low growth rate of HEV-3 at the maternal-fetal interface. Therefore, we had to use lower amounts in order to compare the two genotype progeny virions.

5/ Different scales are used in the bar graphs of decidua and placenta (eg Fig 2a/b), which makes comparison

difficult, for the same cytokine an identical scale should be used e.g. IL-6.

Response:

This has now been mended and identical scales are included.

Reviewer 3:

Summary of comments from this reviewer: Overall, reviewer 3 acknowledged the importance of addressing the high pathogenicity of HEV genotype1 during pregnancy and its closely related, less pathogenic HEV genotype 3 that does not harm pregnant women and their fetus. He/She highlighted the fact that HEV is difficult to grow in cell cultures. However, the reviewer raised some concerns and recommended further evidence of viral replication that we addressed in this revised version of the manuscript and in this point-by-point responses below.

Response to Reviewer's general comments:

The authors chose to inoculate explants of human placenta and decidua with HEV-1 or -3 samples from patient feces to generate stock virus. It is not clear why they did not try to grow the patient-derived HEV-1 and HEV-3 strains first in cell lines optimized for HEV cultivation like HepG2/C3A or the lung carcinoma line A549. Importantly, they show that increasing HEV RNA titers appear in the supernatants of the inoculated explants suggesting susceptibility of these tissues for both HEV-1 and -3.

Response:

HEV is genetically unstable in cell culture. Both synonymous and non-synonymous mutations have been shown to confer higher replication capacities *in vitro*^{17,30}. Therefore, adaption of HEV-1 and HEV-3 from clinical samples in cell lines may inevitably result in the outgrowth of mutated viruses that are different from the original ones found in patient samples. Furthermore, under normal conditions, *in vitro* HEV-1 replication is not trivial and requires the induction of ER stress³¹ that may impair cell integrity. Within this notion, to mimic clinical settings, we deliberately used HEV-1 and HEV-3 strains that represent the clinical agents without any propagation *in vitro*. In contrast to HEV-3, we found that HEV-1 hardly replicates in HEPG2/C3A cells. These results are now included in Supplementary Fig. 3 to support our choice.

The main point is that HEV-1 seems to grow better than HEV-3 in these tissues, which could explain why HEV-1 is more pathogenic for pregnant women and the fetus. The question remains whether the rather moderate differences in the growth rate by a factor of 3-4 between HEV-1 and -3 is sufficient to explain the drastic differences in real patients.

Response:

By using a model relevant for human pregnancy, our data reveals that HEV-1 replicates better than HEV-3 both in tissue explants and primary cells of the maternal-fetal interface, whilst HEV-3 grows much better in hepatocellular carcinoma HEPG2/C3A cells.

We fully agree with the reviewer that the moderate difference in the growth rate by a factor of 3-4 between HEV-1 and -3 is not sufficient by itself to explain the drastically different outcome in pregnant women. However, the difference in viral growth rate was clearly associated with prominent tissue damage in HEV-1 infected samples. Furthermore, viral replication was correlated with impairment of the cytokine balance at the maternal-fetal interface. Similar changes have been reported in pregnancy complications including miscarriage, pre-eclampsia and IUGR¹¹⁻¹⁵. This notion is further supported by experiments in Fig. 4 using UV-inactivated conditioned media. Independently of viral replication, the induction of tissue damage by CM - albeit to a lesser extent than replicative virus - is much higher with HEV-1 than with HEV-3. Interestingly, previous reports have suggested that active HEV replication in placenta in conjunction with other factors contribute to the severity of the disease during pregnancy⁵. These showed that even a moderate difference in the viral growth rate might result in a HEV-1-associated severe pregnancy outcome. As a whole, our data suggest that the increased replicative capacity of HEV-1 associated with the cytokine storm would work in concert to inflict such a drastic outcome of the pregnancy. These clarifications are now included in the revised manuscript (Lines 298-322).

The proof that HEV really grows in these tissues is incomplete (point 1b) because the titer of progeny virus is much lower than the amount of input virus. Thus, additional evidence for replication is needed.

Response:

We would like to thank the reviewer for this valuable suggestion that we undertook. Consequently, a new set of experiments has been performed to provide solid evidence of active viral replication. We applied the following steps to consolidate the data:

- RNAscope® Technology for *in situ* hybridization using a set of probes that cover the whole HEV genome. This analysis clearly demonstrates that HEV-1 replicates much better than HEV-3 at the maternal-fetal interface, be it *decidua basalis* or placenta.

- Inhibition studies using ribavirin as suggested by reviewer 1. The data show clearly a decrease of viral replication in the presence of 50 µM RBV.

These results, now reported in lines 91-107 of the revised manuscript as well as Fig. 1c,d and Supplementary Fig. 1, exclude a passive release of membrane bound particles over time.

The extensive testing of cytokines and other cellular factors changed by the addition of HEV is interesting but insufficient to corroborate conclusions concerning HEV replication.

Response:

We apologize if our statements concerning this matter were not formulated in a clear enough manner. In fact, the extensive cytokine testing was not designed to investigate whether soluble factors impact viral replication, in order to explain the difference in viral growth rate between HEV genotypes. Rather, to provide some insights into the differential pathogenesis of HEV-1 and HEV-3 during pregnancy. More precisely, the cytokine experiments aimed at assessing whether both virus replication and the changes in the milieu balance could be linked to tissue damage.

Nevertheless, we complemented our original data by several experiments addressing the genotype-specific discrepancy in viral replication and included the results in the revised manuscript (Fig. 5, lines 181-204). Since type III IFNs have emerged as major players in the immunity to viral infection at the maternal-fetal interface²⁹, we first quantified IFN-λ1 and IFN-λ2/3 in supernatants from infected cultures. By contrast to HEV-3, HEV-1 infection is associated with a significant decrease in IFN-λ1 and IFN-λ2/3 secretions. This prompted us to assess whether this decrease may impact HEV-1 and HEV-3 replication. Therefore, we analyzed viral replication in decidual and placental tissues in the presence of recombinant IFN-λ1 or IFN-λ2, and found that HEV-1 infection of the decidual explants was prevented by the presence of IFN-λ1 or IFN-λ2, whilst the infection of the fetal placenta was decreased by IFN-λ2 treatment. Taken together, our findings suggest that HEV-1 efficient viral replication may be due to a decrease in interferon type III secretion at the maternal-fetal interface.

Major points:

1. Fig 1.

a. What was the multiplicity of infection, i. e. the copy number per cell number? It is not clear to how many cells and in which volume the inoculum was given.

Response:

Homogenous tissue explant of standard size lead to $2 \times 10^6 \pm 5 \times 10^5$ of isolated cells after enzymatic digestion. Therefore, the estimated MOI is 10. Tissue explants are infected in a 24 well plate (Falcon) in a total volume of 500 µl of medium containing 2% fetal calf serum. This information is now included in M&M (Lines 375-390).

b. From the figures one could guess that the maximum amount of the newly formed virus was ca. 8 times less than the input virus. Thus, it cannot be excluded that the virus detected was transiently bound to the cells and then released without real replication. Detection of minus strand RNA or of subgenomic RNA would be a better proof of replication.

Response:

Extensive washing after tissue contact was strictly applied for initial propagation. In addition, to quantify viral replication in the culture supernatants at different time points post-infection, we harvested 50% of cell culture supernatants and replenished with fresh culture media at each kinetic time-point. Under these experimental conditions, we observed an increase in viral RNA within culture supernatants. Therefore, it is very unlikely that the measured virus results from a passive release of cell membrane bound virions. To further confirm this notion, we probed the viral genome in placental and decidual explants using ISH and analyzed the control of HEV replication by ribavirin (50 μ M) as detailed in the response to reviewer 2's general comments. Taken together, these results provide additional evidence that HEV-1, and to a lesser extent HEV-3, replicates in decidual and placental tissues (Lines 91-107 of the revised manuscript as well as Fig. 1c,d and Supplementary Fig. 1).

c. Does day 0 p. i. mean the time immediately after inoculation of 2x10E7/mL HEV RNA copies and thorough washing? Was there really no HEV RNA detectable in view of a detection limit of 100/mL? Differences in histopathology occurred mostly in syncytiotrophoblasts (HEV-1) and secondarily in villus cores (HEV-3) from 6 donor tissues at 5 days post-infection. Broken syncytiotrophoblasts can result from handling and is atypical of viral infection. How many breaks were observed per explant and was there an increase over time? For HEV-3, by what mechanism did necrotic zones appear in villus cores? What does fold-change mean in pathology of infected tissues relative to controls? Blebbing of syncytiotrophoblasts occurred in controls, HEV-1 and -3. What is the difference?

Response:

Tissue contact with the inoculum is carried out for 24h as stated in the M&M. Explants are then transferred to a new 6 well plate and washed 5 times in 5ml PBS with five changes to new plastic plates. The explants are then laid on top of collagen sponges¹ in the presence of 1ml of complete culture medium. Supernatant day 0 is collected at this step (after infection and washing but before culturing). Residual values from day 0 were 1x10⁵ copies/ml and 3x10⁵ copies /ml for decidua and placenta respectively, regardless of the genotype. These technical details are now included in the M&M section (Lines 385-390). Since these basal values do not change the observed differences between HEV-1 and HEV-3 replication kinetics, we subtracted them from each time point in order to have the most accurate and just representation of viral replication. However, if the reviewer feels that this should be shown, we could modify the graph in Fig. 1 accordingly and show the baseline or add it as a supplementary figure.

Regarding the histopathology experiments, both the syncytial layer and the villous core show signs of necrosis in HEV-1 infected tissues, with significantly higher damage than in the case of HEV-3. We agree with the reviewer that the original Fig. 1c (renamed Fig. 2c,d in the revised version of the manuscript) might appear confusing. To further support this issue, we performed additional experiments using TUNEL assay. Staining of DNA fragmentation showed that HEV-1 infection harms the syncytiotrophoblast layer as well as the villous core. These effects were observed to a lesser degree with HEV-3. These results are shown in Fig. 2a,b and described in lines 110-122 of the revised manuscript.

Since the samples were handled in a similar manner and control explant showed only minor damage and TUNEL staining, it is very unlikely that the harm observed in infected tissue comes from bad handling. Furthermore, damage can be observed as early as three days post-infection but reached significance after 5 days. Our results, therefore, rather support viral involvement or host-related factors in tissue injury. Necrosis and apoptosis were previously reported in animal models and liver biopsies from HEV infected patients^{16,18-21}. The origin of this tissue injury remains elusive and has not been clearly identified as either viral and/or host factors can be incriminated. Nevertheless, mitochondrial damage and activation of the caspase family might underlie tissue injury²¹

In order to quantify the injury associated with HEV-1 and HEV-3 infection, we represented the data as a fold change relative to mock-infected tissue. Of note, necrosis and apoptosis are very low in mock-infected tissue (See Fig. 2 mock-infected tissues).

Whether changes in the syncytiotrophoblast layer can be considered as blebbing or not, is unclear. However, similar changes were observed in placenta obtained from HCMV infected pregnancies³² as well as placenta infected with ZIKV^{33,34}. Furthermore, similar syncytiotrophoblast changes have been observed in pre-eclampsia placenta³⁵.

2. L93. “privileged replication sites” means that there are other sites of replication, but in this part of the text no other sites are mentioned. Since the tissue tropism is the major point of the paper, the authors should re-organize the text to show another tissue or cell culture for comparison as they did later.

Response:

The term “privileged” was used to highlight the advantageous replication of HEV-1 over HEV-3. However, to avoid confusion for the reader, the sentence has been reworded as follows: “Taken together, these experiments demonstrate that decidual and placental tissues support a better replication of HEV-1 compared to HEV-3” (Lines 106-107).

In the revised manuscript we also included a Supplementary Fig. 3, showing the replication kinetics of HEV-1 and HEV-3 in the HepG2/C3A cell line, as well as the intracellular staining of the ORF-2 protein. We found that, by contrast to HEV-3, HEV-1 barely replicates in the HepG2-C3A cell line. These data further confirm the higher tropism of HEV-1 at the maternal-fetal interface.

Fig. 3.

a. The legend should mention at which time point of infection the various cultures were analysed.

Response:

Cytokines and viral RNA were analyzed in the same culture supernatant at 48h post-infection. The obtained values were then used for correlation analysis. This information has been added to the figure legend of the revised manuscript.

b. The panel for IL-6 does not suggest that the virus production has an influence on IL-6 secretion even if the p-value is 0.04. See also lines 124, 125.

Response:

We agree that for the placenta explants, the changes in IL-6 secretion are much lower than that observed for the decidua explants and do not support virus influence on its secretion. Of note when analyzing primary isolated placental stroma cells, changes in IL-6 secretion between HEV-1 and HEV-3 are better characterized. This point has been revised accordingly.

c. L134-141. The UV-treated conditioned medium still contains HEV RNA which by itself may induce cytokines. Can this effect be excluded?

Response:

Our data show here that soluble mediators within the CM can induce tissue injury in the absence of replication. However, we cannot exclude that cells sense viral proteins and/or RNA contained in the UV-treated conditioned media further amplifying the soluble factor release. This feedback loop, might contribute to the observed tissue damage. This point is now discussed in the revised version (Lines 298-322).

4.Fig. 4b. What was the multiplicity of infection for the stromal cells? See point 1b.

Response:

Prior to infection, primary stromal cells derived from either the decidua or placenta, were seeded in 6-well plates overnight (5×10^5 cells /well). Cells were then infected with 10^7 RNA copies of either HEV-1, HEV-3 in a total volume of 1000 μ l of medium containing 2% fetal calf serum. Accordingly, the estimated MOI is 20 for primary isolated cells and HepG2C3A. This information has been added to the M&M section.

5.Fig. 4c.

a.The number of HEV capsid producing stromal cells seems to be low. A quantitative counting of positive cells per total cell number would be desirable.

Response:

We assessed the positive cells per total cell number as follows: we counted the anti-ORF-2-stained cells in ten microscopic overlapping fields (Objective x40) and included the data as a bar graph in the revised Fig. 6.

b.Is this staining pattern really typical for an HEV infection?

c.A convincing positive and negative control is missing, e.g. infected and uninfected HepG2/C3A cells.

Response:

Similar infection rates and staining profiles were observed (0.5%–1.0%) when using the anti-ORF-2 antibodies^{36,37}. The immunostaining profiles were further confirmed in the HepG2C3A cells that were infected for 7 days (Supplementary Fig. 3).

d.The appearance of decidual and placental stromal cells is similar. Usually decidual stromal cells are cultured under conditions that reflect the uterine origin and could effect virus production, which was not done here, but would be relevant to infection HEV during pregnancy.

Response:

The DMEM/F12 culture medium supplemented with fetal bovine serum both preserves the architecture of the maternal-fetal interface and promotes the growth, development and viability of cells. This culture medium has been commonly used by several groups including ourself. Furthermore, these culturing conditions have been also successfully used in the context of other viral infection studies including those investigating hCMV, HIV Zika virus and toxoplasma gondii^{1,33,34,38,39}. Although other culturing media might reflect uterine origin, it might not ensure the benefits of the used medium. Moreover, using the same culture media for all cell and tissue culture facilitate results comparison. We therefore chose to use this culture medium in all our experiments.

6.Fig. 6

a.What was the multiplicity of infection?

Response:

We reiterate our response to point 4 above, the estimated MOI is 20 for primary cells and HepG2C3A.

b.Testing the infectivity in new decidual or placental explants would have been more convincing to prove the differential infectivity.

Response:

The discrepancy in HEV-1 and HEV-3 replication was addressed in tissue explants. The foremost question of supernatant transfer on primary cells and HepG2/3A was to check whether viral replication within maternal-fetal interface give rise to infectious virions. More importantly, we wanted to determine whether newly synthesized virions preserve the observed efficiency and tropism of initial HEV-1 strain. Such information cannot be drawn by using tissue explants instead of primary cells and HepG2/3A.

7.Methods. The calibration of the qPCR for HEV RNA is not sufficiently described. How were the internal standards for HEV RNA generated and calibrated? How does the number of HEV RNA molecules relate to HEV particles and infectious virions?

Response:

We thank the reviewer for raising this point. This has now been included in the revised M&M section. The RNA standards were designed as follows: a fragment within the ORF3 gene (70 nt) was amplified from an HEV genotype 3 infected patient samples. The resulting cDNAs was cloned into pGEM.3Z. The fragment was

then retro-transcribed using T7 RNA polymerase. The obtained positive strand was used as the RNA standard in the quantitative RT-PCR experiments. A standard curve was generated from the serial 10-fold dilutions of this RNA standard. The RNA quantification was performed in a one-step, real-time RT-PCR using a LightCycler 480 instrument (Roche Diagnostics, France) and the following primers, targeting the ORF2/ORF3 overlapping region: the forward primer HEVORF3-S (5'-GGTGGTTTCTGGGGTGAC-3'), the reverse primer HEVORF3-AS (5'-AGGGGTTGGTTGGATGAA-3') and the probe 5'-6-carboxyfluorescein (FAM)-TGATTCTCAGCCCTTCGC-6-carboxytetramethylrhodamine (TAMRA)-3'. The amplification efficiency was then calculated using the standard curve. The detection limit was 100 copies/ml. This method is accredited ISO15189^{4,40,41}. The median values in the stools was 820 TCID50 per million HEV RNA copies (range 650–910). The number of infectious virions in the feces was estimated at 1 in 1000 HEV particles.

Minor points:

d. Spell out abbreviations CPTP, and UCSF, CHU, IFB in the affiliations.

Response:

All abbreviations are now spelled out in the revised manuscript.

e. L31 and later. Replace “non-pathogenic” by “less pathogenic”. HEV-3 is pathogenic for ca. 0.1 % of the infected subjects causing symptomatic acute or chronic hepatitis.

Response:

The term ‘non-pathogenic’ has duly been replaced by the term ‘less-pathogenic’ throughout the text.

f.L44. A recent reference to the current taxonomy of the Hepeviridae would be useful.

Response:

We now include the recent taxonomy from the ICTV Consortium⁴²
Purdy MA, Harrison TJ, Jameel S, Meng XJ, Okamoto H, Van der Poel WHM, Smith DB, Ictv Report Consortium. J Gen Virol. 2017 Nov;98(11):2645-2646. doi: 10.1099/jgv.0.000940. Epub 2017 Oct 12.

g.L82. The gestational age of the explants should be mentioned because they change considerably during pregnancy.

Response:

All explants were prepared from samples of elective pregnancy termination with a gestational age of 8-12 weeks. This information is provided in lines 354-355 of the revised manuscript.

h.L83. It ought to be mentioned here and not only in the methods that the virus came from feces of patients with symptomatic (?) hepatitis E.

Response:

This notion was added in the text and the M&M section (see lines 367-374).

i.Fig.6 Typo: placental

Response:

The typo error has been corrected in the legends of the original Fig. 6 that is now Fig. 8 of the revised manuscript.

References:

- 1 El Costa, H. *et al.* The local environment orchestrates mucosal decidual macrophage differentiation and substantially inhibits HIV-1 replication. *Mucosal Immunol* **9**, 634-646, doi:10.1038/mi.2015.87 (2016).
- 2 Allweiss, L. *et al.* Human liver chimeric mice as a new model of chronic hepatitis E virus infection and preclinical drug evaluation. *J Hepatol* **64**, 1033-1040, doi:10.1016/j.jhep.2016.01.011 (2016).
- 3 van de Garde, M. D. *et al.* Hepatitis E Virus (HEV) Genotype 3 Infection of Human Liver Chimeric Mice as a Model for Chronic HEV Infection. *J Virol* **90**, 4394-4401, doi:10.1128/JVI.00114-16 (2016).
- 4 Chapuy-Regaud, S. *et al.* Characterization of the lipid envelope of exosome encapsulated HEV particles protected from the immune response. *Biochimie* **141**, 70-79, doi:10.1016/j.biochi.2017.05.003 (2017).
- 5 Bose, P. D. *et al.* Evidence of extrahepatic replication of hepatitis E virus in human placenta. *J Gen Virol* **95**, 1266-1271, doi:10.1099/vir.0.063602-0 (2014).
- 6 Moffett, A. & Loke, C. Immunology of placentation in eutherian mammals. *Nat Rev Immunol* **6**, 584-594, doi:10.1038/nri1897 (2006).
- 7 Schmidt, A., Morales-Prieto, D. M., Pastuschek, J., Frohlich, K. & Markert, U. R. Only humans have human placentas: molecular differences between mice and humans. *J Reprod Immunol* **108**, 65-71, doi:10.1016/j.jri.2015.03.001 (2015).
- 8 Tsarev, S. A. *et al.* Experimental hepatitis E in pregnant rhesus monkeys: failure to transmit hepatitis E virus (HEV) to offspring and evidence of naturally acquired antibodies to HEV. *J Infect Dis* **172**, 31-37 (1995).
- 9 Jabrane-Ferrat, N. & Siewiera, J. The up side of decidual natural killer cells: new developments in immunology of pregnancy. *Immunology* **141**, 490-497, doi:10.1111/imm.12218 (2014).
- 10 Siewiera, J. *et al.* Natural cytotoxicity receptor splice variants orchestrate the distinct functions of human natural killer cell subtypes. *Nat Commun* **6**, 10183, doi:10.1038/ncomms10183 (2015).
- 11 Pinheiro, M. B. *et al.* Severe preeclampsia goes along with a cytokine network disturbance towards a systemic inflammatory state. *Cytokine* **62**, 165-173, doi:10.1016/j.cyto.2013.02.027 (2013).
- 12 Saito, S., Nakashima, A., Shima, T. & Ito, M. Th1/Th2/Th17 and regulatory T-cell paradigm in pregnancy. *Am J Reprod Immunol* **63**, 601-610, doi:10.1111/j.1600-0897.2010.00852.x (2010).
- 13 Du, M. R., Wang, S. C. & Li, D. J. The integrative roles of chemokines at the maternal-fetal interface in early pregnancy. *Cell Mol Immunol* **11**, 438-448, doi:10.1038/cmi.2014.68 (2014).
- 14 Lash, G. E. & Ernerudh, J. Decidual cytokines and pregnancy complications: focus on spontaneous miscarriage. *J Reprod Immunol* **108**, 83-89, doi:10.1016/j.jri.2015.02.003 (2015).
- 15 Sharma, S., Godbole, G. & Modi, D. Decidual Control of Trophoblast Invasion. *Am J Reprod Immunol* **75**, 341-350, doi:10.1111/aji.12466 (2016).
- 16 Kamar, N., Dalton, H. R., Abravanel, F. & Izopet, J. Hepatitis E virus infection. *Clin Microbiol Rev* **27**, 116-138, doi:10.1128/CMR.00057-13 (2014).
- 17 Okamoto, H. Culture systems for hepatitis E virus. *J Gastroenterol* **48**, 147-158, doi:10.1007/s00535-012-0682-0 (2013).
- 18 Agrawal, V., Goel, A., Rawat, A., Naik, S. & Aggarwal, R. Histological and immunohistochemical features in fatal acute fulminant hepatitis E. *Indian J Pathol Microbiol* **55**, 22-27, doi:10.4103/0377-4929.94849 (2012).
- 19 Lenggenhager, D. & Weber, A. An Update on the Clinicopathologic Features and Pathologic Diagnosis of Hepatitis E in Liver Specimens. *Adv Anat Pathol* **25**, 273-281, doi:10.1097/PAP.000000000000195 (2018).
- 20 Drebber, U. *et al.* Hepatitis E in liver biopsies from patients with acute hepatitis of clinically unexplained origin. *Front Physiol* **4**, 351, doi:10.3389/fphys.2013.00351 (2013).
- 21 Soomro, M. H. *et al.* Antigen detection and apoptosis in Mongolian gerbil's kidney experimentally intraperitoneally infected by swine hepatitis E virus. *Virus Res* **213**, 343-352, doi:10.1016/j.virusres.2015.12.012 (2016).
- 22 Siewiera, J. *et al.* Human cytomegalovirus infection elicits new decidual natural killer cell effector functions. *PLoS Pathog* **9**, e1003257, doi:10.1371/journal.ppat.1003257 (2013).

- 23 El Costa, H. *et al.* Critical and differential roles of NKp46- and NKp30-activating receptors expressed
by uterine NK cells in early pregnancy. *J Immunol* **181**, 3009-3017 (2008).
- 24 van de Garde, M. D. B. *et al.* Interferon-alpha treatment rapidly clears Hepatitis E virus infection in
humanized mice. *Sci Rep* **7**, 8267, doi:10.1038/s41598-017-07434-y (2017).
- 25 Xu, L. *et al.* RIG-I is a key antiviral interferon-stimulated gene against hepatitis E virus regardless of
interferon production. *Hepatology* **65**, 1823-1839, doi:10.1002/hep.29105 (2017).
- 26 Yin, X. *et al.* Hepatitis E virus persists in the presence of a type III interferon response. *PLoS Pathog*
13, e1006417, doi:10.1371/journal.ppat.1006417 (2017).
- 27 Sayed, I. M. *et al.* Study of hepatitis E virus infection of genotype 1 and 3 in mice with humanised
liver. *Gut*, doi:10.1136/gutjnl-2015-311109 (2016).
- 28 Devhare, P. B., Desai, S. & Lole, K. S. Innate immune responses in human hepatocyte-derived cell
lines alter genotype 1 hepatitis E virus replication efficiencies. *Sci Rep* **6**, 26827,
doi:10.1038/srep26827 (2016).
- 29 Corry, J., Arora, N., Good, C. A., Sadovsky, Y. & Coyne, C. B. Organotypic models of type III
interferon-mediated protection from Zika virus infections at the maternal-fetal interface. *Proc Natl
Acad Sci U S A* **114**, 9433-9438, doi:10.1073/pnas.1707513114 (2017).
- 30 Nagashima, S. *et al.* Analysis of adaptive mutations selected during the consecutive passages of
hepatitis E virus produced from an infectious cDNA clone. *Virus Res* **223**, 170-180,
doi:10.1016/j.virusres.2016.07.011 (2016).
- 31 Nair, V. P. *et al.* Endoplasmic Reticulum Stress Induced Synthesis of a Novel Viral Factor Mediates
Efficient Replication of Genotype-1 Hepatitis E Virus. *PLoS Pathog* **12**, e1005521,
doi:10.1371/journal.ppat.1005521 (2016).
- 32 Tabata, T., Pettitt, M., Fang-Hoover, J., Zydek, M. & Pereira, L. Persistent Cytomegalovirus Infection
in Amniotic Membranes of the Human Placenta. *Am J Pathol* **186**, 2970-2986,
doi:10.1016/j.ajpath.2016.07.016 (2016).
- 33 El Costa, H. *et al.* ZIKA virus reveals broad tissue and cell tropism during the first trimester of
pregnancy. *Sci Rep* **6**, 35296, doi:10.1038/srep35296 (2016).
- 34 Weisblum, Y. *et al.* Zika Virus Infects Early- and Midgestation Human Maternal Decidual Tissues,
Inducing Distinct Innate Tissue Responses in the Maternal-Fetal Interface. *J Virol* **91**,
doi:10.1128/JVI.01905-16 (2017).
- 35 Salgado, S. S. & Salgado, M. K. R. Structural changes in pre-eclamptic and eclamptic placentas--an
ultrastructural study. *J Coll Physicians Surg Pak* **21**, 482-486, doi:08.2011/JCPSP.482486 (2011).
- 36 Riddell, M. A., Li, F. & Anderson, D. A. Identification of immunodominant and conformational
epitopes in the capsid protein of hepatitis E virus by using monoclonal antibodies. *J Virol* **74**, 8011-
8017 (2000).
- 37 Wu, X. *et al.* Pan-Genotype Hepatitis E Virus Replication in Stem Cell-Derived Hepatocellular
Systems. *Gastroenterology* **154**, 663-674 e667, doi:10.1053/j.gastro.2017.10.041 (2018).
- 38 Robbins, J. R., Zeldovich, V. B., Poukchanski, A., Boothroyd, J. C. & Bakardjiev, A. I. Tissue
barriers of the human placenta to infection with *Toxoplasma gondii*. *Infect Immun* **80**, 418-428,
doi:10.1128/IAI.05899-11 (2012).
- 39 Tabata, T. *et al.* Zika Virus Targets Different Primary Human Placental Cells, Suggesting Two Routes
for Vertical Transmission. *Cell Host Microbe* **20**, 155-166, doi:10.1016/j.chom.2016.07.002 (2016).
- 40 Abravanel, F. *et al.* Genotype 3 diversity and quantification of hepatitis E virus RNA. *J Clin
Microbiol* **50**, 897-902, doi:10.1128/JCM.05942-11 (2012).
- 41 Kamar, N. *et al.* Ribavirin for chronic hepatitis E virus infection in transplant recipients. *N Engl J
Med* **370**, 1111-1120, doi:10.1056/NEJMoal215246 (2014).
- 42 Purdy, M. A. *et al.* ICTV Virus Taxonomy Profile: Hepeviridae. *J Gen Virol* **98**, 2645-2646,
doi:10.1099/jgv.0.000940 (2017).

Reviewers' Comments:

Reviewer #1:

None

Reviewer #2:

Remarks to the Author:

In their revised paper "Genotype Specific Pathogenicity of Hepatitis E Virus at the Human Maternal-Fetal Interface", Gouilly et al addressed some of the remarks raised during the initial review process. They performed additional experiments and added clarifications which have contributed to the readability of the paper. Specifically they performed in situ-hybridization experiments showing that around 2% vs 1% of cells are HEV gt 1 and 3 RNA positive respectively in both decidua and placenta tissues. They added ribavirin treatment immediately after inoculation to show slower expansion of HEV gt1 and gt 3 viral loads compared to untreated conditions. In addition, they examined the effect of adding high doses of IFN lambda to the inoculated cultures, inspired by a decrease of IFN -lambda 2/3 levels in the supernatant of HEV gt 1 infected decidua and placenta tissues. They performed histological TUNEL assays to illustrate semi-quantitative differences in tissue apoptosis, next to the previous described histological changes. These additional experiments are helpful in understanding the merits and specificities of the tissue explant and primary stromal cells cultures.

The biggest concern is whether this model is representative of what is happening in humans and can help us in dissecting the clinical differences. We understand that the maximum time to maintain these cultures is two weeks for primary cells and 8-10 days for tissue explants, as stated by the authors in their rebuttal and illustrated by the restricted time axes (between 7 and 14 days) in the different figures. In addition, the model is based on explants from first trimester terminated pregnancies. It is well known, as the authors mention in the discussion, that most of the worst outcomes of HEV gt 1 occur late during pregnancy (3rd trimester) and not during the first trimester (Hepatology. 2015 Dec;62(6):1683-96; International Journal of Gynecology and Obstetrics 85 (2004) 240-244; Liv Int 2018 in press doi: 10.1111/liv.13928). As the hormonal, cytokine and chemokine milieu changes significantly during ongoing pregnancy, the relevance of the findings for the pathogenesis of HEV during late pregnancy would require additional studies in third trimester placenta tissues or ideally ex vivo material from infected patients as recently found in France by the same institute of the authors (Emerg Infect Dis 2018 in press. <https://doi.org/10.3201/eid2405.171845>).

Although the authors should be applauded for the effort they have put into characterizing the model, the very short time frame and rather low level of replication above background (equaling 1×10^5 to 3×10^5 copies/mL distracted from the viral loads) limits the generalization of the findings. This also hampers antiviral studies as cultures had to be supplemented with ribavirin and interferon lambda upon inoculation and not at the plateau of replication to demonstrate their antiviral efficacy. It is puzzling why increases in viral loads above background are seen after addition of either high concentrations of RBV (50 μ M, Suppl Fig 1) or IFN-lambda (100 ng/mL, Fig 5)? Can this still be release from the inoculated cells despite washing? Can the true pathogenesis be modelled when cultures are lost within 8 to 10 days? Also the study of only 2 clinical strains showing 2- to 4-fold different viral kinetics is a limitation. Additional strains could corroborate the found differences. Finally, the model at this time does not provide mechanistic insights into why HEV gt1 and gt3 are different. Is it viral epitopes and therefore immune cell related? Or a specific viral protein as suggested by the authors. The observed IFN lambda decrease can be a start of an explanation, but is not the full story.

Reviewer #3:

Remarks to the Author:

This manuscript is highly improved and the authors have been responsive to all of the critiques.

Reviewer #2 (Remarks to the Author):

In their revised paper “Genotype Specific Pathogenicity of Hepatitis E Virus at the Human Maternal-Fetal Interface”, Gouilly et al addressed some of the remarks raised during the initial review process. They performed additional experiments and added clarifications which have contributed to the readability of the paper. Specifically they performed in situ-hybridization experiments showing that around 2% vs 1% of cells are HEV gt 1 and 3 RNA positive respectively in both decidua and placenta tissues. They added ribavirin treatment immediately after inoculation to show slower expansion of HEV gt1 and gt 3 viral loads compared to untreated conditions. In addition, they examined the effect of adding high doses of IFN lambda to the inoculated cultures, inspired by a decrease of IFN –lambda 2/3 levels in the supernatant of HEV gt 1 infected decidua and placenta tissues. They performed histological TUNEL assays to illustrate semi-quantitative differences in tissue apoptosis, next to the previous described histological changes. These additional experiments are helpful in understanding the merits and specificities of the tissue explant and primary stromal cells cultures.

The biggest concern is whether this model is representative of what is happening in humans and can help us in dissecting the clinical differences. We understand that the maximum time to maintain these cultures is two weeks for primary cells and 8-10 days for tissue explants, as stated by the authors in their rebuttal and illustrated by the restricted time axes (between 7 and 14 days) in the different figures. In addition, the model is based on explants from first trimester terminated pregnancies. It is well known, as the authors mention in the discussion, that most of the worst outcomes of HEV gt 1 occur late during pregnancy (3rd trimester) and not during the first trimester (Hepatology. 2015 Dec;62(6):1683-96; International Journal of Gynecology and Obstetrics 85 (2004) 240–244; Liv Int 2018 in press doi: 10.1111/liv.13928). As the hormonal, cytokine and chemokine milieu changes significantly during ongoing pregnancy, the relevance of the findings for the pathogenesis of HEV during late pregnancy would require additional studies in third trimester placenta tissues or ideally ex vivo material from infected patients as recently found in France by the same institute of the authors (Emerg Infect Dis 2018 in press. <https://doi.org/10.3201/eid2405.171845>).

Although the authors should be applauded for the effort they have put into characterizing the model, the very short time frame and rather low level of replication above background (equaling 1x10⁵ to 3x 10⁵ copies/mL distracted from the viral loads) limits the generalization of the findings. This also hampers antiviral studies as cultures had to be supplemented with ribavirin and interferon lambda upon inoculation and not at the plateau of replication to demonstrate their antiviral efficacy. It is puzzling why increases in viral loads above background are seen after addition of either high concentrations of RBV (50 µM, Suppl Fig 1) or IFN-lambda (100 ng/mL, Fig 5)? Can this still be release from the inoculated cells despite washing? Can the true pathogenesis be modelled when cultures are lost within 8 to 10 days? Also the study of only 2 clinical strains showing 2- to 4-fold different viral kinetics is a limitation. Additional strains could corroborate the found differences. Finally, the model at this time does not provide mechanistic insights into why HEV gt1 and gt3 are different. Is it viral epitopes and therefore immune cell related? Or a specific viral protein as suggested by the authors. The observed IFN lambda decrease can be a start of an explanation, but is not the full story.

Author Response:

We would like to thank the reviewer for appreciating our additional experiments conducted to reinforce and clarify the message of the manuscript. He/she acknowledged that our additional experiments are

highly relevant and improve our understanding of HEV pathogenicity during pregnancy. However, he/she raised additional concerns that we would like to address in this point-by-point response.

Query 1:

The biggest concern is whether this model is representative of what is happening in humans and can help us in dissecting the clinical differences. We understand that the maximum time to maintain these cultures is two weeks for primary cells and 8-10 days for tissue explants, as stated by the authors in their rebuttal and illustrated by the restricted time axes (between 7 and 14 days) in the different figures.

Response 1:

We appreciate the reviewer's understanding that one limitation of the study is inherent to the use of human samples and the fact that the primary tissues cannot be maintained *ex vivo* for a long period of time. However, it is important to highlight that our study represents the first report of HEV *ex vivo* pathogenicity using human maternal and placental tissues and primary cells derived from these tissues. We and others have provided the proof of concept that this model is of high clinical relevance to investigate mechanisms of human congenital infections (HCMV, ZIKV, HIV, Listeria and Plasmodium). Furthermore, this model has already been recommended as relevant for screening immune sera from vaccine trials, passive immune therapies as well as potential anti-viral compounds that can be safely used during pregnancy (Petit et al., 2017 Current Opinion in Virology, doi: 10.1016/j.coviro.2017.11.008) further emphasizing its validity, interest, and usefulness to investigate HEV pathogenicity in manner closest to *in vivo* human clinical settings.

Query 2:

In addition, the model is based on explants from first trimester terminated pregnancies. It is well known, as the authors mention in the discussion, that most of the worst outcomes of HEV gt 1 occur late during pregnancy (3rd trimester) and not during the first trimester (Hepatology. 2015 Dec;62(6):1683-96; International Journal of Gynecology and Obstetrics 85 (2004) 240–244; Liv Int 2018 in press doi: 10.1111/liv.13928). As the hormonal, cytokine and chemokine milieu changes significantly during ongoing pregnancy, the relevance of the findings for the pathogenesis of HEV during late pregnancy would require additional studies in third trimester placenta tissues or ideally *ex vivo* material from infected patients as recently found in France by the same institute of the authors (Emerg Infect Dis 2018 in press. <https://doi.org/10.3201/eid2405.171845>).

Response 2:

We are certainly aware of the case report study by our group in press concerning a chronic HEV-3 in immunocompromised pregnant women. However, placenta collection is subjected to patient consent and it is clearly stated in the pointed-out manuscript that the placenta was not available, which is clearly a measure of burden. Beyond availability and despite the interest of investigating this placenta, it is important to point out that the patient was immunocompromised receiving a combination therapy of Infliximab and Azathioprine. Such a treatment would by itself render the analysis of the placenta irrelevant to our topic. Nevertheless, clinical investigations did not show any HEV RNA or IgM either in the cord blood or the new-born's plasma. We totally agree with the reviewer that further investigations are certainly warranted to fully understand the HEV-1 pathogenesis in term pregnancy. However, we firmly believe that such a study is beyond the scope of our manuscript. Furthermore, our study is the first to address HEV pathogenicity during pregnancy using human tissue explants rather than immortalized or tumoral cell lines. Nevertheless, we discussed the need of further investigation using term placenta to reach a better understanding of the HEV pathogenicity in the discussion section (see paragraph in lines 334-345 of the revised manuscript).

Query 3:

Although the authors should be applauded for the effort they have put into characterizing the model, the very short time frame and rather low level of replication above background (equaling 1×10^5 to 3×10^5 copies/mL distracted from the viral loads) limits the generalization of the findings.

Response 3:

We thank the reviewer for his comment.

Concerning the viral load, we decided to take into account the residual virus at day zero due to the branched architecture of the placental tissue. To the best of our knowledge, most of the studies on HEV have used stem cell progenitors or immortalized cell lines rather than tissue explants. The high residual viral RNA at day 0, compared to other cell line studies, is most likely inherent to the tissue architecture. This is further highlighted by the fact that in primary cells or HepG2/C3A cell line, the amount at day 0 is much lower in the range of 10^3 power. Despite this higher starting point, which by definition lower the replication above background, our model recapitulates the difference in HEV genotype-dependent pathogenicity and allows us to examine the specific anti-viral effect of Ribavirin or IFN- λ . Thus, although our experimental model has its limitations as any other model, the observed tissue damage evaluated through necrosis and apoptosis, combined with the dysregulation in the maternal-fetal secretome within human settings, reinforce the relevance of our study model and findings. For further the reader' appreciation, our concerns about the limitations of the studied model were added to the revised manuscript (see paragraph in lines 334-345 of the revised manuscript).

Query 4:

This also hampers antiviral studies as cultures had to be supplemented with ribavirin and interferon lambda upon inoculation and not at the plateau of replication to demonstrate their antiviral efficacy. It is puzzling why increases in viral loads above background are seen after addition of either high concentrations of RBV (50 μ M, Suppl Fig 1) or IFN-lambda (100 ng/mL, Fig 5)? Can this still be release from the inoculated cells despite washing?

Response 4:

While this might seem quite puzzling, numerous studies have used similar or even much higher amounts of either ribavirin (RBV) or IFN- λ to inhibit HEV replication in various cell models. For instance, in the study by the Steinmann's group, 100 μ M of RBV was able to inhibit HEV replication in the JEG and JAR choriocarcinoma cell lines. Even if the inhibition was quite efficient, the authors still observed some residual viral replication (Knegendorf et al, 2018, doi: 10.1002/hep4.1138).

Regarding the IFN- λ , a recent study by the Feng's group (Yin et al. 2017, doi: 10.1371/journal.ppat.1006417) showed that HEV-3 infection of either HepG2 cells or primary hepatocytes results in the production of IFN- λ (100-200 pg/mL). Their analysis of IFN- λ dose-response effects showed optimal inhibition of 40% with 10 ng/ml. Higher doses of IFN- λ did not improve their observed effect.

Since we are addressing viral replication in the whole tissue organ cultures, we decided to use the optimal doses of 100 ng/mL for IFN- λ and 50 μ M of RVB. Similar to published reports, it is not surprising to observe some residual replication with both molecules at these concentrations. Therefore, we do believe that the residual amount of viral RNA in the presence of inhibitory molecules is due to a partial blockade of viral replication rather than passive release from the inoculum. This is further supported by fact that we do not observe this residual when using UV-irradiated virus (data not shown). This point is discussed in the revised manuscript lines 93-98 for RBV and was already reported in the lines 287-289 for IFN- λ .

Query 5:

Can the true pathogenesis be modelled when cultures are lost within 8 to 10 days? Also the study of only 2 clinical strains showing 2- to 4-fold different viral kinetics is a limitation. Additional strains could corroborate the found differences.

Response 5:

Our actual study as well as several other reports on placental vulnerability to TORCH pathogens coming from the mother have shown that the placental damage is quite rapid and occurs within few days. Since the damage to the placenta and the dysregulation of the microenvironment occur within five days, the model proposed here is highly relevant to study events that occur upon encountering the virus. Early stages of placental development are key for pregnancy success, therefore, even minor damage at any stage can lead to a disastrous outcome of the pregnancy.

We agree with the reviewer, that additional strains are needed to provide a more generalize statement about HEV-1 severity, which would be probably addressed in our futures studies since at this point we do not have access to any other strains. Nonetheless, our data are clearly in agreement with the observed pathogenicity that happens *in utero*. This issue is now mentioned in the discussion (see paragraph in lines 334-345 of the revised manuscript).

Query 6:

Finally, the model at this time does not provide mechanistic insights into why HEV gt1 and gt3 are different. Is it viral epitopes and therefore immune cell related? Or a specific viral protein as suggested by the authors. The observed IFN lambda decrease can be a start of an explanation, but is not the full story.

Response 6:

While we acknowledge the limitations of the study, we think that our work provides valuable insights into the mechanistic that govern HEV-1 pathogenesis in pregnant women. Our study represents the first report of HEV *ex vivo* pathogenicity using human maternal and placental tissues and primary cells derived from these tissues. Furthermore, our data suggest that both virus and host-related factors can contribute to HEV-1 pathogenicity during pregnancy. This issue was discussed in the manuscript (lines 280-326 and lines 343-349 and Fig. 9) and, appears in lines 279-325 of the revised manuscript.

Reviewer #3 (Remarks to the Author):

This manuscript is highly improved and the authors have been responsive to all of the critiques.

Response:

We would like to thank reviewer 3 for his/her original comments and for his positive feedback. We also would like to thank reviewer 1 for his/her original comments.